# DECIPHeR v1: Dynamic fluxEs and ConnectIvity for Predictions of HydRology

Gemma Coxon[1,2], Jim Freer[1,2], Rosanna Lane[1], Toby Dunne[1], Wouter J. M. Knoben[3],
Nicholas J. K. Howden[2,3], Niall Quinn[4], Thorsten Wagener[2,3], Ross Woods[2,3]

[1]Geographical Sciences, University of Bristol, Bristol, United Kingdom, BS8 1SS
[2]Cabot Institute, University of Bristol, Bristol, United Kingdom, BS8 1UJ
[3]Department of Civil Engineering, University of Bristol, Bristol, United Kingdom, BS8 1TR
[4]Fathom Global, The Engine Shed, Station Approach, Bristol, United Kingdom, BS1 6QH

*Correspondence to:* Gemma Coxon (gemma.coxon@bristol.ac.uk)

**Abstract.** This paper presents DECIPHeR (Dynamic fluxEs and ConnectIvity for Predictions of HydRology); a new model framework that simulates and predicts hydrologic flows from spatial scales of small headwater catchments to entire continents. DECIPHeR can be adapted to specific hydrologic settings and to different levels of data availability. It is a flexible model framework which includes the capability to (1) change its representation of spatial variability and hydrologic connectivity by implementing hydrological response units in any configuration, and (2) test different hypotheses of catchment behaviour by altering the model equations and parameters in different parts of the landscape. It has an automated build function that allows rapid set-up across large model domains and is open source to help researchers and/or practitioners use the model. DECIPHeR is applied across Great Britain to demonstrate the model framework. It is evaluated against daily flow time series from 1,366 gauges for four evaluation metrics to provide a benchmark of model performance. Results show the model performs well across a range of catchment characteristics but particularly in wetter catchments in the West and North of Great Britain. Future model developments will focus on adding modules to DECIPHeR to improve the representation of groundwater dynamics and human influences.

# 1    Introduction

Water resources require careful management to ensure adequate potable and industrial supply, to support the economic and recreational value of water, and to minimise the impacts of hydrological extremes such as droughts and floods on the economy, river ecosystems and
human life.  Robust simulations and predictions of river flows are increasingly needed across multiple temporal and spatial scales to support such management strategies (Wagener et al., 2010) that may range from the assessment of local field-scale flood mitigation measures to emerging water challenges at regional to continental scales (Archfield et al., 2015). Such approaches are particularly important, indeed mandated, given national and international
policies on water management, such as the European Union's Water Framework Directive (EC, 2000) and Floods Directive (EC, 2007).  Specifically (inter)national information on water resources, low and high flows is needed to underpin robust environmental management and policy decisions. This requires the effective integration of field observations and numerical modelling tools to provide tailored outputs at gauged and ungauged locations
across a wide range of scales relevant to policy makers and societal needs.

To address this need, a fundamental challenge for hydrologic sciences is to develop hydrological models that represent the complex drivers of catchment behaviour, such as space- and time- varying climate, land cover, human influence etc. (Blöschl and Sivapalan, 1995).  The hydrologic community has made substantial investments to develop and apply
hydrological models over the past 50 years to produce simulations and predictions of surface and groundwater flows, evaporation and soil moisture storage across multiple scales.  These include gridded approaches (e.g. PCR-GLOBWB, (Wada et al., 2014); VIC, (Hamman et al., 2018; Liang et al., 1994); Grid-to-Grid, (Bell et al., 2007); Multiscale Hydrologic Model (Samaniego et al., 2010); DK-model, (Henriksen et al., 2003)), semi-distributed approaches
that aggregate the landscape into hydrologic response units or sub-catchments (e.g. HYPE, (Lindström et al., 2010); SWAT, (Arnold et al., 1998); Topnet, (Clark et al., 2008a)) and many conceptual models applied at the catchment scale (Beven and Kirkby, 1979; Burnash, 1995; Coron et al., 2017; Leavesley et al., 1996; Lindström et al., 1997; Zhao, 1984).  The current generation of hydrological models can represent a range of natural and anthropogenic
processes and various levels of spatial complexity.  Furthermore, there are significant ongoing efforts to represent spatial heterogeneity at finer scales over national-global scales (Bierkens et al., 2015; Wood et al., 2011) and build multi-model frameworks, to test competing hypotheses of catchment behaviour, such as FUSE (Clark et al., 2008a) and SUPERFLEX (Fenicia et al., 2011; Kavetski and Fenicia, 2011).

However, whilst these models have provided a wealth of useful insights and relevant outputs, they either tend to: have a fixed representation of spatial variability (i.e. a single spatial resolution or a single spatial structure such as raster based); lack spatial connectivity between hillslope-to-hillslope and hillslope and riverine components; be computationally expensive; and/or employ a single model structure across the model domain or nested catchment scale.
This impacts our ability to apply models to a wide range of scales, places and water challenges, as different model representations of hydrological processes (i.e. model structure, parameterisations, hydrologic connectivity or spatial variability) are needed to capture heterogeneous hydrological responses and changing landscape connectivity, particularly for local conditions.  Consequently, there is a pressing need to develop new spatially flexible
modelling tools that can be applied to a range of space- and time- scales, and that are based on general hydrological principles applicable to a broad spectrum of different catchment types.  The need for such approaches is well documented in the literature (Clark et al., 2011,

2015; Mendoza et al., 2015) with calls for flexible hydrological modelling systems that can: (1) incorporate different model structures and parameterisations in different parts of the landscape to represent a variety of processes; (2) change their spatial complexity, variability and/or hydrologic connectivity for hillslope elements and river network reaches (Beven and Freer, 2001; Mendoza et al., 2015); and, (3) be applied across a wide range of spatial and temporal scales, and across places (Blöschl et al, 2013). However, few such models exist.

In line with these requirements, we have created a new model framework, DECIPHeR (Dynamic fluxEs and ConnectIvity for Predictions of HydRology), to simulate and predict hydrologic flows and connectivity from spatial scales of small headwater catchments to entire continents. The flexible modelling framework allows users to test different spatial resolutions, spatial configurations (i.e. gridded, semi-distributed or lumped), levels of hydrologic connectivity (i.e. representations of the lateral fluxes of water across model elements) and process representation (i.e. model structure and parameters). DECIPHeR has an automated build function that allows rapid set-up across required model domains with limited user input. The underlying code has been optimised to run large ensembles and enable model uncertainty to be fully explored. This is particularly important given inherent uncertainties in hydro-climatic datasets (Coxon et al., 2015; McMillan et al., 2012) and their impact on model calibration, regionalisation and evaluation (Freer et al., 2004; Kavetski et al., 2006; Kuczera et al., 2010; McMillan et al., 2010, 2011; Westerberg et al., 2016). We have specifically made the model code readable, reusable and open source to allow the broader community to learn from, verify and advance the work described here (Buytaert et al., 2008; Hutton et al., 2016).

In this paper, we (1) describe the key capabilities and concepts that underpin DECIPHeR; (2) provide a detailed discussion of the model code and components; (3) demonstrate its application at the national scale to 1,366 catchments in Great Britain (GB); and, (4) discuss potential future model developments.

## 2    The DECIPHeR Modelling Framework

### 2.1    Key Concepts

The DECIPHeR modelling framework is based on the key concepts enshrined in Dynamic TOPMODEL originally introduced by Beven and Freer, (2001). Since its original development, Dynamic TOPMODEL has been applied in a wide range of studies (Freer et al., 2004; Liu et al., 2009, p.200; Metcalfe et al., 2017; Page et al., 2007; Younger et al., 2008) and integrated into other modelling frameworks (e.g. HydroBlocks, Chaney et al., 2016). The core ideas of Dynamic TOPMODEL were three-fold (Beven and Freer, 2001); 1) to allow more flexibility in the definition of similarity in function for different points in the landscape, 2) to implement a non-linear routing of subsurface flow that simulates dynamically variable upslope subsurface contributing area and 3) to remain computationally efficient so that uncertainty in hydrological simulations can be estimated.

To realise this, Dynamic TOPMODEL uses hydrological response units (HRUs) to group raster-based information into non-contiguous spatial elements in the landscape that share similar characteristics (see Figure 1). Each HRU maintains hydrological connectivity in the landscape via weightings that determine the proportions of lateral subsurface flux from each HRU to all connected HRUs and flows to river cells. This solution offers key advantages in capability to traditional grid-based or lumped approaches employed by many hydrological models. Firstly, the user can split up the catchment using, for example, different landscape attributes (e.g. geology, land use) and/or spatially varying inputs (e.g. rainfall, evaporation,

etc.) to define spatial similarity. This capability allows the user to modify the spatial complexity, resolution and/or hydrologic connectivity of hillslope elements and river network reaches in any configuration. Secondly, each HRU is treated as a separate functional unit in the model which can have different process conceptualisations and parameterisations. This
means that more process complexity can be incorporated where needed to better suit local conditions (e.g. to account for 'point-source' human influences or more complex hydrological processes such as surface-groundwater exchanges). Finally, by grouping together similar parts of the landscape, HRUs minimise run times of the model compared to grid-based or fully distributed formulations, while still allowing model simulations to be
mapped back into space.

While these key concepts that underpin Dynamic TOPMODEL address many of the challenges outlined in the introduction section, for the most part the model has only ever been applied to a single catchment or very simple nested catchments in headwater basins (Peters et al., 2003). Consequently, we have completely restructured and rewritten the model code and
added several new features to improve the flexibility and automation of the original Dynamic TOPMODEL code so the model can be applied from single small headwater catchments to regional, national and continental scales. These changes include:

1. Both legacy and new model code has been updated to a FORTRAN 2003 compliant version with new array and memory handling to allow significantly larger and more
20        complex gauging networks to be processed
2. The model build process is now fully automated to allow national/continental scale data to be easily and quickly processed, and to build and apply models in complex multi-catchment regions.
3. New model code and functions have been written to:
25        a. Enable greater flexibility in the complexity and spatial characteristics of river network and routing properties. A newly developed river network scheme allows flow simulations to be produced for any gauged or ungauged point on a river network and segment river reaches into any length for individual hillslope-river flux contributions.
30        b. Ensure that multiple points on the river network can be initialised via local storages and fluxes in each HRU successfully.
          c. Seamlessly facilitate DTA classification layers and results into rainfall-runoff model configuration that allows each individual HRU to have a different model structure, parameters, and climatic inputs.
4. A new analytical solution of the subsurface flow equations has been implemented, resulting in increased computational speed and numerical stability
      5. The model can be easily adopted and adapted because it is open source, version controlled and includes a detailed user manual

HRUs are defined prior to rainfall-runoff modelling and DECIPHeR consists of two key steps where (1) digital terrain analyses are performed to define the gauge network, set up the river network and routing, discretise the catchment into HRUs and characterise the spatial variability and hydrologic connectivity in the landscape, and (2) HRUs are run in the rainfall runoff model to provide flow timeseries. These two steps are described in the following
sections. More detailed descriptions of the input and output files, code workflows and codes can be found in the user manual.

## 2.2 Digital Terrain Analysis (DTA)

The DTA in DECIPHeR constructs the spatial topology of the model components to define hillslope and riverine elements. The DTA defines the spatial extent of every HRU based upon multiple attributes, quantifies the connectivity between these HRU's in the landscape, determines the river network and all downstream routing properties, and determines the extent and where simulated output variables (i.e. discharge) should be produced (including gauged or ungauged locations) (see Figure 1).

### 2.2.1 Data Prerequisites

The minimum data requirement to run the DTA is a digital elevation model (DEM) and XY locations where flow time series are needed on the river network. The DEM must contain no sinks or flat areas to ensure that the river network and catchments can be properly delineated as is common in digital terrain analyses. This means that any real inland sinks (such as lakes) will be filled. Accounting for these features in the modelling framework will be a focus for future model development.

Additional data can also be incorporated depending on data availability and modelling objectives. A river network can be supplied if the user wishes to specify headwater cells from a predefined river network and reference catchment areas and masks can be used to identify the best station location on the river network. Depending on user requirements, topographic, land use, geology, soils, anthropogenic and climate attributes can be supplied to define the spatial topology and thus differences in model inputs, structure and parameterisation.

### 2.2.2 River Network, Catchment Identification and River Routing

DECIPHeR generates streamflow estimates at any point on the river network specified by the user. A river network is generated in DECIPHeR which matches the DEM flow direction and always connects to the boundary of the DEM or the sea. The river network is created from a list of headwater cells, which the functions can use/produce in three different ways depending on user requirements and/or data availability:

1. A list of pre-defined headwater (i.e. starting) river locations read into the DTA algorithms from a file
2. Headwater cells are found from a pre-defined river network
3. Where no pre-defined river network or headwater locations are available, then headwater cells are found from a river network which is derived from cells that meet thresholds of accumulated area and/or topographic index

Each headwater location is then routed downstream in a single flow direction via the steepest slope until reaching a sea outlet, other river or edge of the DEM, to construct a contiguous river network for the whole area of interest. Gauge locations are then generated on the river network from the point locations specified by the user. If a reference catchment mask or area is available, catchment masks are produced for candidate river cells found in a given radius and the catchment mask with the best fit to the reference mask or area is chosen as the gauge location. Otherwise the closest river cell is chosen as the gauge location.

Catchment masks are created from the final gauge list, with both individual masks for all the points specified on the river network and a combined catchment mask with the nested catchment masks created for use in the creation of the hydrological response units. From the river network and gauge locations, the river network connectivity is derived with each river section labelled with a unique river ID. A suite of routing tables is also produced so that each

ID knows its downstream connections and to allow multiple routing schemes to be configured (see section 2.3.4 for a description of the current routing scheme implemented in the modelling framework). These codes also provide the option of setting a river reach length where output time series can also be specified at different reach lengths between gauges (see Figure 1, HRU Setup D).

### 2.2.3 Topographic Analysis

Topography, slope, accumulated area and topographic index are important properties of the landscape to aid the definition of hydrologic similarity and more dominant flow pathways. In DECIPHeR, they provide the basis for river routing and river network configuration and they also can be used to help determine the initial separation of landscape elements for defining hydrological similarity using percentiles of accumulated area, elevation and slope (in addition to alternative catchment attributes such as urban extent, geology, landuse, soils etc.).

Topographic index is calculated using the M8 multiple flow directional algorithm of (Quinn et al., 1995). The DTA calculates slope, accumulated area and topographic index for the whole domain. It uses the river mask to define the cells where accumulated area cannot accumulate downstream and the catchment mask to ensure accumulated area does not accumulate across nested catchment boundaries.

### 2.2.4 Hydrological Response Units

The most critical aspect of running DECIPHeR is to define HRU's according to user requirements. The HRU configuration determines the spatial connectivity and complexity of model conceptualisation as well as the spatial variability of inputs and conceptual structure and parameters to be implemented in each part of the landscape. Any number of different spatial discretisations can be derived and subsequently applied in the DECIPHeR framework allowing the user to experiment with different model structures and parameterisations and modify representations of spatial variability and hydrologic connectivity.

In the DTA, hydrologically similar points in the landscape are grouped together so that each HRU is a unique combination of four different classification layers. These specify: (1) the initial separation of landscape elements from topographic information (e.g. slope, accumulated area and/or elevation); (2) inputs; (3) process conceptualisations; and (4) parameters implemented for each HRU store in the model (see Figure 2). These classification layers can be derived from climatic inputs, such as spatially varying rainfall and potential evapotranspiration, and landscape attributes such as geology, land use, anthropogenic impacts, soils data, slope, accumulated area. The simplest setup will consist of one HRU per catchment while the most complex can consist of one HRU for every grid cell (i.e. fully distributed).

To maintain hydrological connectivity in the landscape, the proportions of flow between the cells comprising each HRU are calculated based on accumulated area and slope. The flow fractions are then aggregated into a flow distribution matrix that summarises the proportions (weightings) of lateral subsurface flow from each HRU either to (1) itself, (2) another HRU or (3) a river reach. For $n$ hydrological response units, the weights ($W$) are defined as:

$$W = \begin{pmatrix} w_{1,1} & \cdots & w_{1,n} \\ \vdots & \ddots & \vdots \\ w_{n,1} & \cdots & w_{n,n} \end{pmatrix}$$

**Equation 1**

Where each row defines how the HRU's output is distributed to other HRU's, any river reaches or itself and each column represents the total input to each HRU at every time step as the weighted sum of all the upstream outputs. Each row and column sum to one to ensure mass balance. The weights are detailed in a HRU flux file (which is fixed for a simulation) as a flow distribution matrix along with tabulated HRU attributes to provide information on which inputs, parameter and model structure type each HRU is using.

## 2.3    Rainfall-Runoff Modelling

### 2.3.1    Data Pre-requisites

To run the rainfall-runoff modelling component of DECIPHeR, time series forcing data of rainfall and potential evapotranspiration are required. Discharge data can also be provided for gauged locations and are used to initialise the model.

Besides forcing data, the model also needs, (1) the HRU flux file and routing files produced by the DTA, (2) a parameter file specifying parameter bounds for Monte-Carlo sampling of parameters and (3) project/settings files specifying the number of parameter sets to run, which HRU and input file to use etc.

### 2.3.2    Initialisation

Initialisation is an important step for any rainfall-runoff model. To ensure that subsurface flows, storages and the river discharge have all stabilised can be particularly problematic when modelling regionally over a large area as not all HRU's will initialise at the same rate (depending on size and slope characteristics).

A simple homogenous initialisation is currently implemented in DECIPHeR where the storage deficits for all HRU's are determined from an initial discharge. This is calculated as a mean area weighted discharge of the starting flows at timestep 1 for all output points on the river network. If a gauge does not have an initial flow, then the initial flow is either calculated from the mean of the data or set to a value of 1 mm/day (as a representative starting flow for most catchments) if no flow data is available. The initial discharge is assumed to be solely due to the subsurface drainage into the river so is used as the starting value for $Q_{SAT}$ (subsurface flow) and to determine the associated storage and unsaturated zone fluxes. The model is then run for an initialisation period to allow its model stores and fluxes to fully stabilise with the catchment climatic information. Initialisation periods depend in part on the parameterisation of the model simulation run as well as the size and characteristics of the catchment being considered.

### 2.3.3    Parameters

DECIPHeR can be run either using default parameter values or through Monte-Carlo sampling of parameters between set parameter bounds to produce ensembles of river flows. In the DTA, the user can set different parameter bounds for each HRU or sub-catchment thus specifying areas of the landscape where different parameter bounds may be needed. Alternatively, a single set of parameter bounds can be applied across the model domain.

For the model structure provided in the standard build and described below, there are seven parameters that can be sampled or set to default parameters. These parameters describe the transmissivity of the subsurface, the water holding capacity and permeability of soils and the channel routing velocity (see Table 1). More parameters can easily be added by the user if required for different model structures by changing the model source code.

### 2.3.4  Model Structure

The description below details the model structure that is provided in the open source code (see Figure 3 and Table 1). While the code is built to be modular and extensible so that a user can easily implement multiple model structures if so wished, the aim of this paper and the initial focus of the code development was on applying the model across large scales and beginning with a release that has relatively simple representations of the core processes. Thus, we provide a single model structure in the open source code that serves as a model benchmark to be built upon in future iterations.

The model structure consists of three stores defining the soil profile ($S_{RZ}$, $S_{UZ}$, $S_D$ in Figure 3), which are implemented as lumped stores for each HRU. The first store is the root zone storage ($S_{RZ}$). Precipitation ($P$) is added to this store and then evapotranspiration ($ET$) is calculated and removed directly from the root zone. The maximum specific storage of $S_{RZ}$ is determined by the parameter $SR_{max}$. Actual evapotranspiration from each HRU depends on the potential evapotranspiration ($PET$) rate supplied by the user and the root zone storage using a simple common formulation where evapotranspiration is removed at the full potential rate from saturated areas (i.e. if the root zone storage is full) and at a rate proportional to the root zone storage in unsaturated areas:

$$ET = PET * (S_{RZ}/SR_{max})$$

**Equation 2**

Once the root zone reaches maximum capacity (i.e. deficit of zero and conceptually analogous to field capacity), any excess rainfall input is added to the unsaturated zone ($S_{uz}$) where it is routed to the saturated zone ($S_D$). If the saturated zone is also full (as determined by $S_{max}$), $Q_{EXUS}$ is added to the saturation excess storage ($S_{EX}$) and routed directly overland as saturated excess overland flow ($Q_{OF}$). The unsaturated zone links the $S_{RZ}$ and saturated zones according to a linear function that includes a gravity drainage time delay parameter ($T_d$) for vertical routing through the unsaturated zone. The drainage flux ($Q_{uz}$) from the unsaturated zone to the saturated zone is at a rate proportional to the ratio of unsaturated zone storage ($S_{uz}$) to storage deficit ($S_D$):

$$Q_{UZ} = S_{UZ}/(S_D * T_d)$$

**Equation 3**

Changes to storage deficits for each HRU are dependent on recharge from $S_{UZ}$ ($Q_{UZ}$), fluxes from upslope HRUs ($Q_{IN}$) and downslope flow out of each HRU ($Q_{SAT}$), with subsurface flows for each HRU distributed according to the DTA flow distribution matrix described in section 2.2.4.

$$\frac{dS_D}{dt} = Q_{SAT} - Q_{IN} - Q_{UZ}$$

**Equation 4**

Where $S_D$ is the current deficit in the saturated zone, $Q_{SAT}$ is outflow from this HRU, $Q_{IN}$ is inflow into the HRU representing subsurface flow from other HRUs and $Q_{UZ}$ is inflow into the HRU representing drainage from the unsaturated zone of this HRU. This equation is solved sequentially for each HRU and provides values for the deficit $S_D$ and outflow $Q_{SAT}$ at time step *t* for each HRU. In DECIPHeR, this equation is solved analytically (see appendix for derivation of this solution), assuming a transmissivity profile that declines exponentially

with depth and is truncated at depth $S_{max}$ such that no flow is generated when the deficit is greater than $S_{max}$ (Beven and Freer, 2001). The analytical solution provides better computational speed and increased numerical stability compared to the iterative 4-point numerical scheme described by Beven and Freer (2001).

The exponential transmissivity profile takes the shape (Beven and Freer, 2001; eq. 6):

$$Q_{SAT} = T_0 \tan \beta \exp(-fz) = Q_0 exp(-\bar{S}/SZM)$$

**Equation 5**

The truncated exponential transmissivity profile takes the shape (rewritten from Beven and Freer, 2001; eq. 9):

$$Q_{SAT} = \begin{cases} Q_0 \cos \beta \left[ exp(-\cos \beta \, \bar{S}/SZM) - exp(-\cos \beta \, S_{max}/SZM) \right] & \bar{S} \leq S_{max} \\ 0 & \bar{S} > S_{max} \end{cases}$$

**Equation 6**

Where $\beta$ is the mean slope of the HRU and $\bar{S}$ is the average deficit across the HRU. The parameter, *SZM,* sets the rate of the exponential decline in saturated zone hydraulic transmissivity with depth thereby controlling the shape of the recession curve in time. The

parameter, $S_{max}$, sets the saturated zone deficit threshold at which downslope flow between HRUs no longer occurs.  If the storage deficit is less than zero (i.e. the soil is at or above its saturation capacity), then excess storage ($Q_{EXS}$) is added to saturation excess overland flow ($Q_{OF}$).  $Q_0$ is the maximum rate of $Q_{SAT}$ from a HRU when the HRU is at saturation and is calculated from:

$$Q_0 = \frac{T_0}{e^\lambda}$$

**Equation 7**

Where the parameter $T_0$ determines the lateral saturated hydraulic transmissivity at the point when the soil is saturated and $\lambda$ is the average topographic index across the HRU.

Channel flow routing in DECIPHeR is modelled using a set of time delay histograms that are

derived from the digital terrain analyses for the points where output is required.  A fixed channel wave velocity (*CHV*) is applied throughout the network to account for delay and attenuation in the simulated flows ($Q_{SIM}$).  DECIPHeR is a mass conserving model and therefore the model water balance always closes (subject to small rounding errors).

## 2.4    Model Implementation

The DECIPHeR model code is available on github (https://github.com/uob-hydrology/DECIPHeR) and is accompanied by a user manual which provides a detailed description of the file formats, how to run the codes and a code workflow.  All the model code is written in FORTRAN for its speed, efficiency and ability to process large scale spatial datasets.  Two additional bash scripts are provided as an example of calling the digital terrain

analysis codes.

## 3    Great Britain National Model Implementation and Evaluation

While the modelling framework has a wide range of functionality, in this paper we wanted to demonstrate the ability of the model to be applied across a large domain to generate

ensembles of flows at thousands of gauging stations and evaluate its current capability across large scales to guide future model developments. Consequently, we applied DECIPHeR to 1,366 gauges in Great Britain (GB) and in this section we describe the model setup, input data, evaluation criteria and model results.

## 3.1   Great Britain Hydrology

Catchments in Great Britain (GB) cover a wide hydrologic and climatic diversity. Hydro-climatic characteristics were derived from rainfall, potential evapotranspiration and flow data described in Section 3.3.1. Figure 4 shows the mean annual rainfall, mean annual potential evapotranspiration, runoff coefficient, and slope of the flow duration curve between the 30 and 70 flow percentiles for the 1,366 catchments in this study. Rainfall is highest in the West and North of GB and lowest in the East and South ranging from 540 to 3400 mm/year (Figure 4a), while potential evapotranspiration is highest in the East and South and lowest in the West and North ranging from 370 to 545 mm/year (Figure 4b). This regional divide of rainfall and potential evapotranspiration is reflected in the runoff coefficients (Figure 4c) where generally runoff coefficients are lowest in the East and South and highest in the North and West. Slope of the flow duration curve (Figure 4d) is a more mixed picture across GB with lower values (i.e. a less variable flow regime) found in North-East Scotland, Midlands and patches of the South-East and higher values (i.e. a more variable flow regime) in the West, with the highest values for ephemeral and/or small streams in the South-East.

River flows vary seasonally with the highest totals generally occurring during the winter months when rainfall totals are highest and evapotranspiration totals are lowest, and the lowest totals during the summer months (April – September) resulting from lower precipitation totals and higher evapotranspiration losses due to seasonal variations in energy inputs. Snowmelt has little impact on river flows in GB except for some catchments in the Scottish Highlands where snowmelt contributions can impact the flows. River flow patterns are also heavily influenced by groundwater contributions from various regional aquifer systems. In catchments overlying the Chalk outcrop in the South-East of the GB, flow is groundwater-dominated with a predominantly seasonal hydrograph that responds less quickly to rainfall events. Land use and human influences also significantly impact river flows, with flows most heavily modified in the South-East and Midland regions of England due to high population densities.

## 3.2   Digital Terran Analyses for GB

To implement DECIPHeR across GB, the UK NEXTMAP 50m gridded digital elevation model was used as the basis of the Digital Terrain Analysis (Intermap, 2009). The first step was to ensure that the DEM contained no sinks or flat areas before being run through the DTA codes. Many freely available packages and codes exist to sink fill DEMs but for use with large national data sets, a two-stage process is often necessary to ensure no flat areas in the DEM and that important features, such as steep sided valleys, are not filled due to pinch points in the DEM. For this study, we first applied an optimised pit removal routine (Soille, (2004), code available on github  https://github.com/crwr/OptimizedPitRemoval). This tool uses a combination of cut and fill to remove all undesired pits while minimizing the net change in landscape elevation. We then applied a sink fill routine to ensure no flat areas remained in the DEM.

The inputs and outputs for the GB DTA is summarised in Figure 5. To build the river network, we first extracted headwater cells from the Ordnance Survey MasterMap Water Network Layer; a dense national river vector dataset for GB. These headwater cells were then routed downstream via the steepest slope to generate the river network used by the

model. This ensures that the DEM and the calculated stream network are consistent for flow accumulations based on surface slope. Locations of 1,366 National River Flow Archive gauges were used to define the gauging network and specify points on the river network where output was required. We used NRFA catchment areas and masks as a reference guide to evaluate the best point for the gauge locations from potential river cell candidates within a local search area. Slope, accumulated area and the topographic index were then calculated for every grid cell and routing files produced.

Finally, we chose three classifiers to demonstrate the modelling framework while ensuring the number of HRUs was still computationally feasible for modelling across a large domain, these being:

1. The catchment boundaries for each gauge were used to ensure minimal fluxes across catchment boundaries.
2. A 5km grid for the rainfall and potential evapotranspiration inputs was used to represent the spatial variability in climatic inputs across GB.
3. Three equal classes of slope and accumulated area were implemented resulting in HRU's that cascade downslope to the valley bottom.

### 3.3 Rainfall Runoff Modelling

### 3.3.1 Input and Evaluation Datasets

Daily data of precipitation, potential evapotranspiration and discharge for a 55-year period from 01/01/1961–31/12/2015 were used to run and assess the model. This period was chosen as an appropriate test for the model covering a range of climatic conditions and to demonstrate the model's ability to simulate long time periods within uncertainty analyses frameworks. The year 1961 was used as a warm-up period for the model; therefore no model evaluation was quantified in this period.

A national gridded rainfall and potential evapo-transpiration product was used as input into the model. Daily rainfall data were obtained from the CEH Gridded Estimates of Areal Rainfall dataset (CEH-GEAR) (Keller et al., 2015; Tanguy et al., 2016). This dataset consists of 1km$^2$ gridded estimates of daily rainfall from 1961 - 2015 for Great Britain and Northern Ireland derived from the Met Office UK rain gauge network. The observed precipitations from the rain gauge network are quality controlled and then natural neighbour interpolation is used to generate the daily rainfall grids. Daily potential evapotranspiration data were obtained from the CEH Climate hydrology and ecology research support system potential evapotranspiration dataset for Great Britain (CHESS-PE) (Robinson et al., 2016). This dataset consists of 1km$^2$ gridded estimates of daily potential evapotranspiration for Great Britain from 1961 - 2015 calculated using the Penman-Monteith equation and data from the CHESS meteorology dataset. Both datasets were aggregated to a 5km grid as forcing for the national model run.

The model was evaluated against daily streamflow data for the 1366 gauges obtained from the National River Flow Archive (www.nrfa.ceh.ac.uk). This data is collected by measuring authorities including the Environment Agency (EA), Natural Resources Wales (NRW) and Scottish Environmental Protection Agency (SEPA) and then quality controlled before being uploaded to the NRFA site.

### 3.3.2   Model Structure and Parameters

To initially evaluate the model, DECIPHeR was run within a Monte-Carlo simulation framework whereby 10000 parameter sets were randomly sampled from a uniform prior distribution. This number of parameter sets was chosen to provide a reasonable sampling of the parameter space for demonstration purposes, however, for a full evaluation of the parameter space, more parameter sets would be needed.

These parameters were applied uniformly across the HRUs and used within a single model structure (as described in Section 2.3.4). Given the wide range of hydroclimatic conditions across GB, sampling of the feasible parameter space was ensured by using wide sampling ranges based on previous studies that have used Dynamic TOPMODEL (Beven and Freer, 2001; Freer et al., 2004; Page et al., 2007) (Table 2).

### 3.3.3   Model Evaluation

Daily time series of discharge for the 10,000 model simulations from each gauge were evaluated against daily observed flow for all 1,366 gauges. This is a challenging test for the model as these catchments cover a large range of hydrologic behaviour across GB and are impacted by a variety of climatic, geological and anthropogenic processes as outlined in Section 3.1. However, evaluating the model over such a large number of gauges acts as a benchmark of model performance and a means of identifying future areas for model development.

To benchmark model performance, we wanted to evaluate the model's ability to capture a range of hydrologic behaviour including maintaining overall water balance, capturing flow variability, reproducing low and high flows and the timing of flows. Consequently, multiple metrics, including hydrological signatures, standard hydrological model performance metrics and statistics of the flow time series were used to provide insights into model performance. Based on previous studies evaluating national scale models (McMillan et al., 2016) and considering a diagnostic approach to model evaluation (Coxon et al., 2014; Gupta et al., 2008; Yilmaz et al., 2008); four metrics were chosen which are summarised in Table 3 alongside their equations i) NSE (Nash and Sutcliffe, 1970), ii) Slope of the Flow Duration Curve (Yadav et al., 2007) iii) Bias in Runoff Ratio (Yilmaz et al., 2008) and iv) Low Flow Volume (Yilmaz et al., 2008).

These metrics are also used to determine a behavioural ensemble of parameter sets. The focus of this model application is to demonstrate the model can be run in a Monte Carlo framework. Consequently, while many different approaches could be used to determine a behavioural ensemble of parameter sets (see for example (Beven, 2006; Coxon et al., 2014; Krueger et al., 2010; Westerberg et al., 2011)), in this study we adopt a simple approach to produce ensembles of flows. The four metrics described above are combined and the behavioural ensemble was then taken as the top 1% of the model simulations according to this combined score. To calculate the combined score, each metric was ranked in turn, these ranks were summed, and all simulations sorted by the total combined rank. Weaker and stricter performance thresholds in NSE and bias metrics were also defined to further explore the performance of the ensembles against a common set of criteria (see Table 3). These were chosen based on previous studies and although subjective, the hydrological modelling community is yet to agree on benchmarks for the comparison of model performance (Seibert et al., 2018).

### 3.4 Results

### 3.4.1 Digital Terrain Analysis and Model Simulation

DECIPHeR was set up for GB covering a total catchment area of 154,763km$^2$ for 1366 gauges and 365 principal basins. Principal basin area ranged from 7.87km$^2$ to 9935km$^2$ with a median of 137km$^2$. Using the HRU classifiers specified in Section 3.2, the number of HRUs contained within each principal basin ranged from 17 to 8978 with a median of 123 HRUs. HRU area ranged from 0.0025km$^2$ to 14.33km$^2$ with a median HRU area of 0.65km$^2$.

In total 13,660,000 55 year time series, flow simulations were produced. One simulation over the 55 year time period for the largest river basin (9935km$^2$) with 8978 HRUs takes approximately 15 minutes to run on a standard CPU, outputting simulated discharge for all the 98 gauges that lie within the Thames at Kingston river basin. For the smallest river basin that has 17 HRUs and one river gauge, a single simulation over the 56 year time period on a standard CPU takes less than a second.

### 3.4.2 Overall Model Performance

Our first assessment of model performance is the overall model performance for the four performance metrics calculated from the 10000 simulated daily flow time series produced for each gauge. Figure 6 shows the percentage of catchments that met the stricter and weaker performance thresholds defined in Table 3 from the entire ensemble of 10000 model simulations and from the top 1% behavioural ensemble generated from the combined ranking of the four metrics. Our results show that most catchments are able to meet both the performance thresholds. The vast majority of catchments (92%) gain a NSE score greater than zero (i.e. better than mean climatology) and 80% of the catchments gain a NSE score greater than 0.5. The model does well in reproducing Low Flow Volumes and Slope of the Flow Duration Curve with most gauges (98 and 96% respectively) meeting the stricter performance threshold.

RRBIAS evaluates the model's ability to reproduce water balance in the catchment; the current implementation of the model has to maintain mass balance while many of the observed flow data for many of these catchments does not maintain mass balance either due to inter-catchment groundwater flows, anthropogenic influences such as surface and ground water abstractions, or data errors (this is further discussed in section 4.4.4). Consequently, RRBIAS is a more difficult metric for the model to capture and this is reflected by the fact that 75% of the catchments meet the weaker threshold and just over 62% meet the stricter threshold.

These numbers decrease slightly for the behavioural ensemble as expected due to trade-offs between the four metrics but the overall trends remain the same.

### 3.4.3 Spatial Model Performance

To analyse model performance spatially across GB, the four evaluation metrics for the best simulation (as defined by the combined rank across all four metrics) for each catchment is summarised in Figure 7.

For NSE, model performance is variable across the country but generally, better model performance is found in the wetter catchments in the North and West of GB, with poorer model performance in drier catchments in the South and East. Model performance is poor in groundwater dominated areas, particularly in the underlying chalk regions in the South East. This region has particularly low runoff coefficients (see Figure 4d) and does not maintain mass balance with large water losses. Consequently, results for RRBIAS shows that the

model tends to over-estimate flows in the South-East. While bias in the runoff ratio shows the model is generally over-estimating flows, biases in the low flow volume is a more mixed picture with the model under-estimating low flows in some locations, particularly in the Midlands and North East Scotland. From Figure 4d, these areas are characterised by particularly low flow duration curve slopes suggesting strongly damped flow responses with high baseflow. Flow in the Midlands region is heavily regulated by reservoirs which sustain low flows and could be a potential reason for over-estimating low flows in this area. The bias in slope of the flow duration curve shows DECIPHeR does well at reproducing the flow variability but tends to under-estimate the slope in Scotland and North Wales suggesting that the hydrographs in these catchments are too smooth and not sufficiently flashy.

### 3.4.4   Relationship Between Model Performance and Catchment Characteristics

To further analyse and understand the reasons for good/poor model performance, relationships between key catchment characteristics and model performance were further explored. Firstly, the catchments were grouped according to key catchment characteristics based on discharge; runoff coefficient and base flow index. The 5th, 50th and 95th percentiles of NSE and RRBIAS were calculated from the ensemble of runs for all catchments within each group to explore relationships between model performance and catchment characteristics (see Table 4). The relationship between runoff coefficient, wetness index and RRBIAS was also analysed to further explore the importance of water gains/losses on model performance.

There is a clear link between model performance and catchments with a low runoff coefficient. Table 4 highlights poor model performance in catchments where observed runoff coefficients are less than 0.2. In this group, the model always over-predicts (as shown by the RRBIAS result) and consequently leads to poor NSE scores. Figure 8 shows that for many catchments where the model over-predicts flows (and particularly for catchments with a runoff coefficient less than 0.2) observed potential evapotranspiration estimates are not high enough to account for water losses culminating in an over-estimation of flows. This is unsurprising given that currently the model maintains water balance and can't lose or gain water beyond the 'natural' conceptualisations of precipitation, discharge and evaporation dynamics. Consequently, we are either missing a process (such as water loss due to inter-catchment groundwater flows or anthropogenic impacts) or the data is wrong.

Poorer model performance is also found in high BFI catchments (Table 4), however, the results also show we can also gain very good simulations in these types of catchments (5th percentile has a NSE score of 0.83), hence the challenge is to better understand water losses/gains in groundwater catchments to improve the representation of groundwater dynamics in the model.

### 3.4.5   Simulated Flow Time Series

Finally, we examined the simulated flow time series for six example catchments with different characteristics. Figure 9 shows the observed discharge, observed precipitation and the 5th-95th percentile uncertainty bounds of the behavioural simulations for six catchments with different characteristics (see Table 5) for a representative two-year period of the 55-year time series simulated. The 5th-95th percentile uncertainty bounds are generated from the likelihood-weighted distribution of the top 1% of the model simulations using the GLUE framework (Beven, 2006).

Our results show the model can capture a range of different hydrological dynamics from wetter catchments in the North-West (Figure 9a) to drier catchments in the South-East

(Figure 9b). While model performance for groundwater catchments can be very good (Figure 9c and Table 5), it also shows that we need to incorporate additional model capability to simulate the dynamics of groundwater dominated catchments. Where we have a very low (for Great Britain) runoff coefficient, this is assumed to involve water losses into a more regional groundwater storage not expressed at the outlet and not yet represented in this version of the model (Figure 9d). While the catchments shown in Figure 9a-d are relatively un-impacted by human influences, the catchments shown in Figure 9e and 9f are heavily impacted by human influences and highlight the challenge of simulating flows nationally across catchments with diverse hydrological behaviour.

## 4    Outlook and Ongoing Developments

### 4.1    National Scale Model Evaluation

This is the first study to comprehensively benchmark hydrological model performance across GB. We calculated four evaluation metrics for 10,000 model simulations for 1,366 GB gauges to provide an initial benchmark of model performance. DECIPHeR generally performs well for the flow time series evaluated in this study, with better results in the West and North in wet catchments as compared to drier catchments in the South and East. This is a common finding for hydrological models, with many studies finding poor model performance and greatest water balance errors in drier catchments (Gosling and Arnell, 2011; McMillan et al., 2016; Newman et al., 2015; Pechlivanidis and Arheimer, 2015). These results are also reflected in other GB model evaluation studies. For example, Coxon et al., (2014) applied FUSE to 24 GB catchments and found the best model performances in wet catchments compared to dry, chalk catchments, (Rudd et al., 2017) evaluated G2G for low flows across 61 GB catchments and found positive bias in low flow volumes in small catchments in the South-East of England and (Crooks et al., 2010) evaluated PDM across 120 GB catchments and found poorer model performance in groundwater dominated, drier catchments.

Poor model performance in these catchments is partially due to some of the metrics chosen in this study, for example percent bias is most sensitive to small absolute biases in the driest catchments when compared to other metrics such as absolute bias. However, positive bias in the runoff ratio could be caused by a number of factors such as under-estimation of potential evapotranspiration (there are other UK gridded potential evapotranspiration products which estimate much higher potential evapotranspiration), inter-catchment groundwater flows, and/or human influences such as water abstraction. Population density is much higher in the South and East compared to the North and West so this regional disparity in model performance could also be explained by a greater rate of abstractions and managed watercourses which alter the flow time series. For example, 55% of the effective rainfall in the Thames catchment is licensed for abstraction (Thames Water, 2017).

These results provide an initial test of DECIPHeR capabilities against a large sample of catchments, but this is only a first-order evaluation of model performance. A more rigorous evaluation would assess the model: over different seasons (Freer et al., 2004); under changing climatic conditions (Fowler et al., 2016); for different hydrological extremes (Coron et al., 2012; Veldkamp et al., 2018; Zaherpour et al., 2018); for multiple objectives simultaneously (Kollat et al., 2012); and, incorporate input and flow data uncertainty (Coxon et al., 2014; Kavetski et al., 2006; McMillan et al., 2010; Westerberg et al., 2016).

### 4.2    Characterising Spatial Heterogeneity and Connectivity

The intended use of DECIPHeR is to determine how much spatial variability and complexity is required for a given set of modelling objectives. It can be run as a lumped model (1 HRU),

semi-distributed (multiple HRUs) or fully gridded (HRU for every single grid cell). In this paper DECIPHeR was applied across 1,366 GB gauges, with catchment masks, 5 km input grids and three classes of accumulated area and slope as classifiers for the hydrological response units, resulting in a total of 133,286 HRUs. Future work needs to consider the appropriate spatial complexity and hydrologic connectivity needed to represent relevant processes (Andréassian et al., 2004; Blöschl and Sivapalan, 1995; Boyle et al., 2001; Chaney et al., 2016; Clark et al., 2015; Metcalfe et al., 2015; Wood et al., 1988). While this work highlights the clear potential of a computationally-efficient large-scale modelling framework that can run large ensembles, a balance is required to ensure computational efficiency when running large ensembles that also maintains sufficient spatial complexity to represent different hydrological processes.

### 4.3 Hypothesis Testing and Model Parameterisation

To demonstrate the modelling framework we implemented a single model structure, provided in the open source model code, in all HRUs across GB and did not experiment with different model structures in different parts of the landscape. This provides a good benchmark of DECIPHeR's ability at the national scale across GB, but the results suggest different model structures are needed to represent a greater heterogeneity of hydrological responses beyond the conceptual dynamics currently implemented in this simple model (as shown in Figure 9). We can gain new process understanding of regional differences in catchment behaviour by testing different model representations (Atkinson et al., 2002; Bai et al., 2009; Perrin et al., 2001). Future work will concentrate on adding modules to DECIPHeR to enhance performance across national and continental scales with a focus on improved representation of groundwater dynamic and human influences to address poor model performance in catchments with a low runoff coefficient. Furthermore, we have ensured the code is open-source and well-documented so that the hydrological community can contribute new/different conceptualisations of the processes shown in this paper.

It is challenging to parameterise a hydrological model across large scales. Here we simply applied the same parameter set across each catchment. Using this basin-by-basin approach has the disadvantage of producing a "patchwork quilt" of parameter fields, with discontinuities in parameter values across catchment boundaries. This is only effective for gauged catchments (Archfield et al., 2015). Ongoing work aims to address these issues by implementing the multiscale parameter regionalisation (MPR) technique for DECIPHeR across GB. This technique links model parameters to geophysical catchment attributes through transfer functions applied at the finest possible resolution (Samaniego et al., 2010). The coefficients of the transfer functions are then calibrated, and parameters are upscaled to produce spatially consistent fields of model parameters at any resolution across the entire model domain. The MPR technique has been applied elsewhere, proving that it can produce seamless parameter fields across large domains and produce scale-invariant parameters (Kumar et al., 2013; Mizukami et al., 2017; Samaniego et al., 2017), which is ideal for a flexible framework such as DECIPHeR.

### 5 Conclusions

DECIPHeR is a new flexible modelling framework which can be applied from small catchment to continental scale for complex river basins resolving small-scale spatial heterogeneity and connectivity. The model is underpinned by a flexible, computationally efficient framework with a number of novel features:

1. *Spatial variability and connectivity* - ability to modify spatial variability and connectivity in the model via the specification of hydrological response units with different topographic, landscape, input layers
2. *Model structures and parameterisations* - ability to experiment with different model structures and parameterisations in different parts of the landscape
3. *Computationally efficient* - grouping of hydrologically similar points in the landscape into hydrological response units enables faster run times
4. *Automated build* – to allow easy application over large scales
5. *Open source* - the open source model code is implemented in Fortran, with a user manual to help researchers and/or practitioners to use the model.

This paper describes the modelling framework and its key components and demonstrates the model's ability to be applied a large model domain. DECIPHeR is shown to be computationally efficient and perform well over large samples of gauges. This work highlights the potential for catchment to continental scale predictions, by making use of available big datasets, advances in flexible modelling frameworks and computing power.

**Appendix A - Analytical Solution for Kinematic Subsurface Flow**

## 1. Introduction

This appendix provides an analytical solution to the equations which were solved numerically by Beven and Freer (2001) in Dynamic TOPMODEL to route subsurface flows. Here, we use calculus to integrate the relevant equations through time, as opposed to the finite volume scheme described by Beven and Freer (2001), for better computational speed and increased numerical stability.

This development starts from the kinematic wave description of flow (a partial differential equation) and integrates that partial differential equation along the flow direction to obtain an ordinary differential equation in time. We then integrate that ordinary differential equation in time to get an analytical solution which gives the flow and storage at the end of each timestep, in relation to the conditions at the start of the timestep, and the inflow from both upslope and from drainage. Each Hydrological Response Unit (HRU) in the model may be comprised of one or more sets of spatially contiguous cells. We use the term "spatial element" (SE) to refer to one of these contiguous sets of cells within a HRU. This is the same scale as referred to by Beven and Freer (2001) as a "group of elements".

By first integrating the kinematic wave equation in space, we have effectively chosen to model the flow at this scale using a nonlinear reservoir, so there is no wave travelling in space within a spatial element. A wave-like behaviour at larger scales is mimicked by having the groups of elements in a type of cascade (linked by the weighting matrix). This approach of integrating in space is the same as selected by Beven and Freer (2001), using their finite volume approach.

## 2. Flow in a Spatial Element

Assume the spatial element (SE) has area A, and that x is distance measured along the flow direction of the SE. Define $Q$ as the downslope flow rate [L$^3$/T] at some point x. Assume that the flow is kinematic, i.e. that $Q$ depends only on $S$ [L], the local storage deficit per unit area, and the SE geometry. The drainage input from above is assumed to be $r$ [L/T]. Assume the width of the SE is $w(x)$, at distance $x$ [L]. At any point $x$ in the SE we can write a partial differential equation for $Q$:

$$\frac{\partial Sw}{\partial t} = \frac{\partial Q}{\partial x} - rw \qquad\qquad \textbf{Equation A1}$$

This is a kinematic wave equation describing the subsurface flow at point x within a SE. Note that both $S$ and $r$ have been multiplied by $w$, the width of the SE, so that they can be compared with $Q$, which is the total flow through the SE, at distance $x$.

To simplify the problem, we will now average over the entire SE, along the flow direction, $x$, from the upslope end ($x$=0) to the downslope end ($x$=L). This will produce an equation describing how $\bar{S}$, the SE-average of $S$, changes with time.

$$\frac{1}{L}\int_0^L \frac{\partial Sw}{\partial t}\,dx = \frac{1}{L}\int_0^L \frac{\partial Q}{\partial x}\,dx - \frac{1}{L}\int_0^L rw\,dx \qquad\qquad \textbf{Equation A2}$$

$$\frac{1}{L}\int_0^L \frac{\partial Sw}{\partial t}\,dx = \frac{Q(L,t)-Q(0,t)}{L} - r\frac{1}{L}\int_0^L w\,dx \qquad\qquad \textbf{Equation A3}$$

The variables $Q(0,t)$ and $Q(L,t)$ refer to flows at the upslope and downslope ends of the SE [L$^3$/T]

If we assume $S$ and $w$ are uncorrelated as $x$ varies, and let $W = \frac{1}{L}\int_0^L w\,dx$ then

$$W \frac{\partial \frac{1}{L}\int_0^L S \mathrm{d}x}{\partial t} = \frac{Q(L,t)-Q(0,t)}{L} - rW \qquad \textbf{Equation A4}$$

Dividing by W,

$$\frac{\partial \bar{S}}{\partial t} = \frac{Q(L,t)-Q(0,t)}{LW} - r \qquad \textbf{Equation A5}$$

Note that $A = LW$ is the area of the SE, so we can now define q=Q/A as flow per unit plan area [L/T] which is the same dimension as used by Beven and Freer (2001).

$$\frac{\partial \bar{S}}{\partial t} = q(L,t) - (q(0,t) + r) \qquad \textbf{Equation A6}$$

In equation 6, $q(0,t)$ and $r$ are assumed to be known, and $q(L,t)$, the outflow from the SE, is assumed to be a function of the mean deficit $\bar{S}$. Thus the SE is being modelled as a nonlinear reservoir, where $\bar{S}$ is the state variable, the input is $q(0,t) + r$, and the outflow $q(L,t) = f(\bar{S}(t))$. Note that the inflow is now assumed to be applied as a spatially uniform flux within the SE, rather than being applied at x=0. There is no representation of motion within the SE. Motion at larger scales is represented by the cascading of flow from one reservoir to another.

Note that in the following equations, Q is equivalent to Q$_{SAT}$ (eq. 4). Because no motion within the SE is represented, Q$_{IN}$ and Q$_{UZ}$ (eq. 4) can be lumped together into a single term, here called r.

**Analytical solutions for an exponential conductivity profile**

There are several parsimonious descriptions of the vertical profile of saturated hydraulic conductivity which are hydrologically plausible. Here we consider the standard exponential profile and a profile truncated at finite depth. In each case we find the analytical solution for both $\bar{S}$ and $q(L,t)$ as functions of time. Analytical solutions are also possible for the parabolic and linear profiles given in Ambroise et al (1996).

Define $u = (q(0,t) + r)$ and $q = q(L,t)$

$$q = q_0 exp(-\bar{S}/m) \qquad \textbf{Equation A7}$$

$$\frac{\partial \bar{S}}{\partial t} = q - u \qquad \textbf{Equation A8}$$

If we substitute 7 into 8, and integrate 8 from $\bar{S}(0)$ at t=0 up to $\bar{S}(t)$, we obtain the intermediate result

$$exp(\bar{S}/m) = \frac{q_0}{u} + exp(\bar{S}(0)/m)\left(1 - \frac{q_0\,exp(-\bar{S}(0)/m)}{u}\right)exp\left(-\frac{ut}{m}\right) \qquad \textbf{Equation A9}$$

From this we can get expressions for both $\bar{S}(t)$ and q(t)

$$\bar{S}(t) = m\,log\left[\frac{q_0}{u} + \left(\frac{1}{exp(-\bar{S}(0)/m)} - \frac{q_0}{u}\right)exp\left(-\frac{ut}{m}\right)\right] \qquad \textbf{Equation A10}$$

$$q(t) = \left[\frac{1}{u} + \left(\frac{1}{q(0)} - \frac{1}{u}\right)exp\left(-\frac{ut}{m}\right)\right]^{-1} \qquad \textbf{Equation A11}$$

In the special case where u=0, we instead obtain

$$\bar{S}(t) = m\,log\left(exp(\bar{S}(0)/m) + \frac{q_0 t}{m}\right) \qquad \textbf{Equation A12}$$

$$q(t) = \left[\frac{1}{q(0)} + \frac{t}{m}\right]^{-1} \qquad \textbf{Equation A13}$$

Note that parameters $m$ and $q_0$ in these equations are equivalent to SZM and $Q_0$ as defined in the paper.

**Exponential truncated smoothly at Smax (Beven and Freer, 2001 equation 9)**

$$q = \begin{cases} q_0 \cos\beta \left[ exp(-\cos\beta\, \bar{S}/m) - exp(-\cos\beta\, S_{max}/m) \right] & \bar{S} \leq S_{max} \\ 0 & \bar{S} > S_{max} \end{cases}$$ **Equation A14**

$$q = \begin{cases} q_1 exp(-\cos\beta\, \bar{S}/m) - q_2 & \bar{S} \leq S_{max} \\ 0 & \bar{S} > S_{max} \end{cases}$$ **Equation A15**

Where $q_1 = q_0 \cos\beta$ and $q_2 = q_0 \cos\beta \exp(-\cos\beta\, S_{max}/m)$

Let's look first at the case where $\bar{S} \leq S_{max}$

$$\frac{\partial \bar{S}}{\partial t} = q_1 exp(-\bar{S}/(m/\cos\beta)) - (q_2 + u)$$ **Equation A16**

If we let $m_2 = m/\cos\beta$ and $u_2 = q_2 + u$, then we can rewrite this as

$$\frac{\partial \bar{S}}{\partial t} = q_1 exp(-\bar{S}/m_2) - u_2$$ **Equation A17**

This is now exactly the same form as the exponential profile above, so the solution is
formally identical: we just put $q_1$ instead of $q_0$, $m_2$ instead of $m$, and $u_2$ instead of $u$. The resulting equations are:

$$\bar{S}(t) = m_2 \log\left[ \frac{q_1}{u_2} + \left( \frac{1}{exp\left(-\frac{\bar{S}(0)}{m_2}\right)} - \frac{q_1}{u_2} \right) exp\left(-\frac{u_2 t}{m_2}\right) \right]$$ **Equation A18**

$$q(t) = \left[ \frac{1}{u_2} + \left( \frac{1}{q(0)} - \frac{1}{u_2} \right) exp\left(-\frac{u_2 t}{m_2}\right) \right]^{-1}$$ **Equation A19**

This solution collapses to the standard exponential result if $\cos\beta = 1$ and $S_{max} = \infty$.
Note that provided $S_{max} < \infty$, then $q_2 > 0$ so $u_2 > 0$, and there is no need to consider the case of zero forcing.

The deficit cannot go beyond $S_{max}$ as a result of outflow; however deficits larger than $S_{max}$ can arise through evaporation (this prepares for future developments, evaporation is not currently included in the conceptualisation of the saturated zone). Here we consider the case
where $\bar{S} > S_{max}$, so q=0, but u>0, so the deficit is decreasing.

$$\frac{\partial \bar{S}}{\partial t} = -u$$ **Equation A20**

This can be integrated to give

$$\bar{S}(t) = \bar{S}(0) - ut$$ **Equation A21**

If $\bar{S}(t) < S_{max}$ then we switch to the $\bar{S} \leq S_{max}$ solution partway through the computational
interval. We use the equation for $\bar{S}$ (t) when $\bar{S} \leq S_{max}$, because in that case the q(t) equation will lead to division by zero if it is started at q(0)=0.

**Extra note on computational issues:**

If $u_2$ is very small but not zero, numerical problems can arise in the calculation of $\bar{S}(t)$, because of loss of significance when subtracting two numbers of very different magnitudes.
This can lead to calculating the logarithm of zero during calculation of $\bar{S}(t)$.

This can be avoided by making a Taylor series expansion of $\bar{S}$ (t) for small non-zero values of $u_2$. We obtain

$$\bar{S}(t) \cong m_2 log \left[ \frac{q_1}{u_2} + \left( \frac{1}{exp\left(-\frac{\bar{S}(0)}{m_2}\right)} - \frac{q_1}{u_2} \right) \left( 1 - \frac{u_2 t}{m_2} + \frac{1}{2!}\left(\frac{u_2 t}{m_2}\right)^2 \right) \right]$$

**Equation A22**

If we expand and then neglect terms in $u_2{}^2$ we obtain

$$\bar{S}(t) \cong m_2 log \left[ \left( \frac{1}{exp\left(-\frac{\bar{S}(0)}{m_2}\right)} + \frac{q_1 t}{m_2} \right) - \frac{u_2 t}{m_2} \left( \frac{1}{exp\left(-\frac{\bar{S}(0)}{m_2}\right)} - \frac{1}{2}\frac{q_1 t}{m_2} \right) \right]$$

**Equation A23**

We use this solution in cases where $\frac{u_2 t}{m_2} \ll 1$, currently implemented as $\frac{u_2 t}{m_2} < 10^{-10}$

## Code Availability

The DECIPHeR model code is open source and freely available under the terms of the GNU General Public License version 3.0. The model code is written in fortran and is provided through a Github repository: https://github.com/uob-hydrology/DECIPHeR.

*Persistent identifier*: https://doi.org/10.5281/zenodo.2604120

## Data Availability

The model forcing and model evaluation datasets used in this paper are publicly available. The CEH-GEAR and CHESS-PE datasets are freely available from CEH's Environmental Information Data Centre and can be accessed through https://doi.org/10.5285/33604ea0-
c238-4488-813d-0ad9ab7c51ca and https://doi.org/10.5285/8baf805d-39ce-4dac-b224-c926ada353b7 respectively. Observed discharge data from the National River Flow Archive is available from the NRFA website. Model output will be made available via CEH's Environmental Information Data Centre.

## Author Contribution

G Coxon and T Dunne wrote and modified the majority of the source code with contributions from R Lane, N Quinn and J Freer. J Freer provided overall oversight for the model development. R Woods and W Knoben derived the analytical solution for the subsurface zone equations. G Coxon prepared the input data and produced and evaluated the model simulations shown in this paper, with input from all co-authors on the experimental design.
The manuscript was prepared by G Coxon with contributions from all co-authors.

## Competing Interests

The authors declare that they have no conflict of interest.

## Acknowledgements

G Coxon, J Freer, T Wagener, R Woods and N Howden were supported by NERC MaRIUS:
Managing the Risks, Impacts and Uncertainties of droughts and water Scarcity, grant number NE/L010399/1. Partial support for T Wagener comes from a Royal Society Wolfson Research Merit Award and for N Howden from a University Research Fellowship. R Lane and W Knoben were funded as part of the Water Informatics Science and Engineering Centre for Doctoral Training (WISE CDT) under a grant from the Engineering and Physical
Sciences Research Council (EPSRC), grant number EP/L016214/1.

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

**Table 1.** Overview of DECIPHeR's stores, fluxes and parameters

| **Stores** | | |
|---|---|---|
| $S_{RZ}$ | Root Zone Storage | m |
| $S_{UZ}$ | Unsaturated Storage | m |
| $S_{EX}$ | Saturation Excess Storage | m |
| $S_D$ | Saturated Storage Deficit | m |
| **Internal Fluxes** | | |
| $Q_{UZ}$ | Drainage Flux | m ts$^{-1}$ |
| $Q_{IN}$ | Upslope Input Flow | m ts$^{-1}$ |
| $Q_{EXS}$ | Saturated Excess Flow | m ts$^{-1}$ |
| $Q_{EXUS}$ | Precipitation Excess Flow | m ts$^{-1}$ |
| $Q_{OF}$ | Overland Flow (sum of $Q_{EXS}$ and $Q_{EXUS}$) | m ts$^{-1}$ |
| $Q_{SAT}$ | Saturated Flow | m ts$^{-1}$ |
| **External Fluxes: Input** | | |
| $P$ | Precipitation | m ts$^{-1}$ |
| $E$ | Potential Evapotranspiration | m ts$^{-1}$ |
| $Q_{obs}$ | Observed Discharge (for starting value of $Q_{SAT}$) | m ts$^{-1}$ |
| **External Fluxes: Output** | | |
| $Q_{sim}$ | Simulated Discharge | m ts$^{-1}$ |
| **Model Parameters** | | |
| $SZM$ | Form of exponential decline in conductivity | m |
| $SR_{max}$ | Maximum root zone storage | m |
| $SR_{init}$ | Initial root zone storage | m |
| $T_d$ | Unsaturated zone time delay | ts m$^{-1}$ |
| $CHV$ | Channel routing velocity | m ts$^{-1}$ |
| $ln(T_0)$ | Lateral saturated transmissivity | ln(m$^2$ ts$^{-1}$) |
| $S_{max}$ | Maximum effective deficit of saturated zone | m |

**Table 2.** Parameter Ranges

| Parameter | Units | Lower Bound | Upper Bound |
|-----------|-------|-------------|-------------|
| $SZM$ | m | 0.001 | 0.15 |
| $SR_{max}$ | m | 0.005 | 0.3 |
| $SR_{init}$ | m | 0 | 0.01 |
| $T_d$ | m hr$^{-1}$ | 0.1 | 40 |
| $CHV$ | m hr$^{-1}$ | 100 | 4000 |
| $ln(T_0)$ | ln(m$^2$ hr$^{-1}$) | -7 | 7 |
| $S_{max}$ | m | 0.3 | 3 |

**Table 3.** Evaluation metrics used in the study

| Evaluation Metric | Equation | Focus | Performance Threshold | |
|---|---|---|---|---|
| | | | Weaker | Stricter |
| Nash Sutcliffe Efficiency | $NSE = 1 - \dfrac{\sum_{i=1}^{n}(QO - QS)^2}{\sum_{i=1}^{n}(QO - \overline{QO})^2}$ | High Flows, Timing | 0 | 0.5 |
| Bias in Runoff Ratio | $RRBias = \dfrac{\sum(QS - QO)}{\sum QO} * 100$ | Water Balance | 20 | 10 |
| Bias in Low Flow Volume | $LFVBias = -100 * \dfrac{\sum_{p=70}^{95}(\log(QS_p) - \log(QO_p))}{\sum_{p=70}^{95}(\log(QO_p))}$ | Low Flows | 20 | 10 |
| Bias in Slope of the Flow Duration Curve between the 30th and 70th percentile | $SFDCBias = \dfrac{[\log(QS_{30}) - \log(QS_{70})] - [\log(QO_{30}) - \log(QO_{70})]}{[\log(QO_{30}) - \log(QO_{70})]} * 100$ | Flow variability | 20 | 10 |

**Table 4.** Summary statistics of DECIPHeR performance metrics for GB with catchments grouped by runoff coefficient and base flow index.  Percentiles are taken from the behavioural ensemble from all catchments within each group.  The column 'N' indicates the number of catchments in each group.  Cells are coloured according to the thresholds outlined in section 4.3.3, green for the stricter threshold, yellow for the weaker threshold and red where it doesn't meet either of the thresholds.

| | Runoff Coefficient | | | | | | | Base Flow Index | | | | | | |
|---|---|---|---|---|---|---|---|---|---|---|---|---|---|---|
| | N | NSE (-) | | | RRBias (%) | | | N | NSE | | | RRBias | | |
| | | 95th | Med | 5th | 95th | Med | 5th | | 95th | Med | 5th | 95th | Med | 5th |
| **0-0.2** | 85 | -73 | -4.4 | 0.35 | 41 | 177 | 894 | 20 | 0.11 | 0.44 | 0.76 | -31 | 0.54 | 134 |
| **0.2-0.4** | 362 | -1.4 | 0.36 | 0.73 | -0.5 | 22 | 123 | 320 | -0.1 | 0.57 | 0.79 | -12 | 1.4 | 100 |
| **0.4-0.6** | 348 | 0.12 | 0.54 | 0.81 | -3.4 | 5.8 | 39 | 629 | -0.1 | 0.54 | 0.80 | -8.9 | 3.9 | 81 |
| **0.6-0.8** | 352 | 0.31 | 0.65 | 0.83 | -10 | 0.14 | 14 | 257 | -1.5 | 0.51 | 0.82 | -10 | 8 | 113 |
| **>0.8** | 219 | 0.02 | 0.64 | 0.81 | -41 | -6 | 3.5 | 140 | -37 | 0.04 | 0.83 | -32 | 31 | 540 |

**Table 5.** Catchment characteristics and model performance for the six catchments shown in Figure 9. Baseflow index is a measure of the proportion of the river runoff that can be classified as baseflow and is derived from Marsh and Hannaford (2008). Water balance is calculated as mean annual rainfall minus mean annual discharge and potential evapotranspiration (as actual evapotranspiration is not available). NSE and BiasRR for the best ranked simulation according to the combined score described in Section 3.3.3 are shown for each catchment alongside the NSE and BiasRR derived from the mean of the behavioural ensemble.

| Gauge Number | River | Gauging Station | Catchment Area (km$^2$) | Mean Annual Rainfall (mm/year) | Mean Annual Potential Evapotranspiration (mm/year) | Mean Annual Discharge (mm/year) | Runoff Coefficient (-) | Water Balance (mm/year) | Baseflow Index (-) | Best Ranked Simulation | | Ensemble Mean | |
|---|---|---|---|---|---|---|---|---|---|---|---|---|---|
| | | | | | | | | | | NSE (-) | RRBias (%) | NSE (-) | RRBias (%) |
| 76014 | Eden | Kirkby Stephen | 69 | 1531 | 453 | 1230 | 0.8 | -152 | 0.26 | 0.77 | -2.6 | 0.79 | -4.9 |
| 37005 | Colne | Lexden | 238 | 582 | 529 | 143 | 0.25 | -91 | 0.52 | 0.63 | 18.8 | 0.43 | 21.3 |
| 43005 | Avon | Amesbury | 324 | 781 | 513 | 352 | 0.45 | -84 | 0.91 | 0.91 | -0.1 | 0.93 | 0.3 |
| 43004 | Bourne | Laverstock | 164 | 800 | 514 | 153 | 0.19 | 133 | 0.91 | <0 | 147.4 | <0 | 148 |
| 25023 | Tees | Cow Green Reservoir | 58 | 1696 | 446 | 1598 | 0.94 | -348 | 0.57 | 0.10 | -8.5 | 0.10 | -12.9 |
| 39001 | Thames | Kingston | 9948 | 724 | 513 | 200 | 0.28 | 11 | 0.65 | 0.56 | 49 | 0.40 | 48.9 |

## Figure Captions

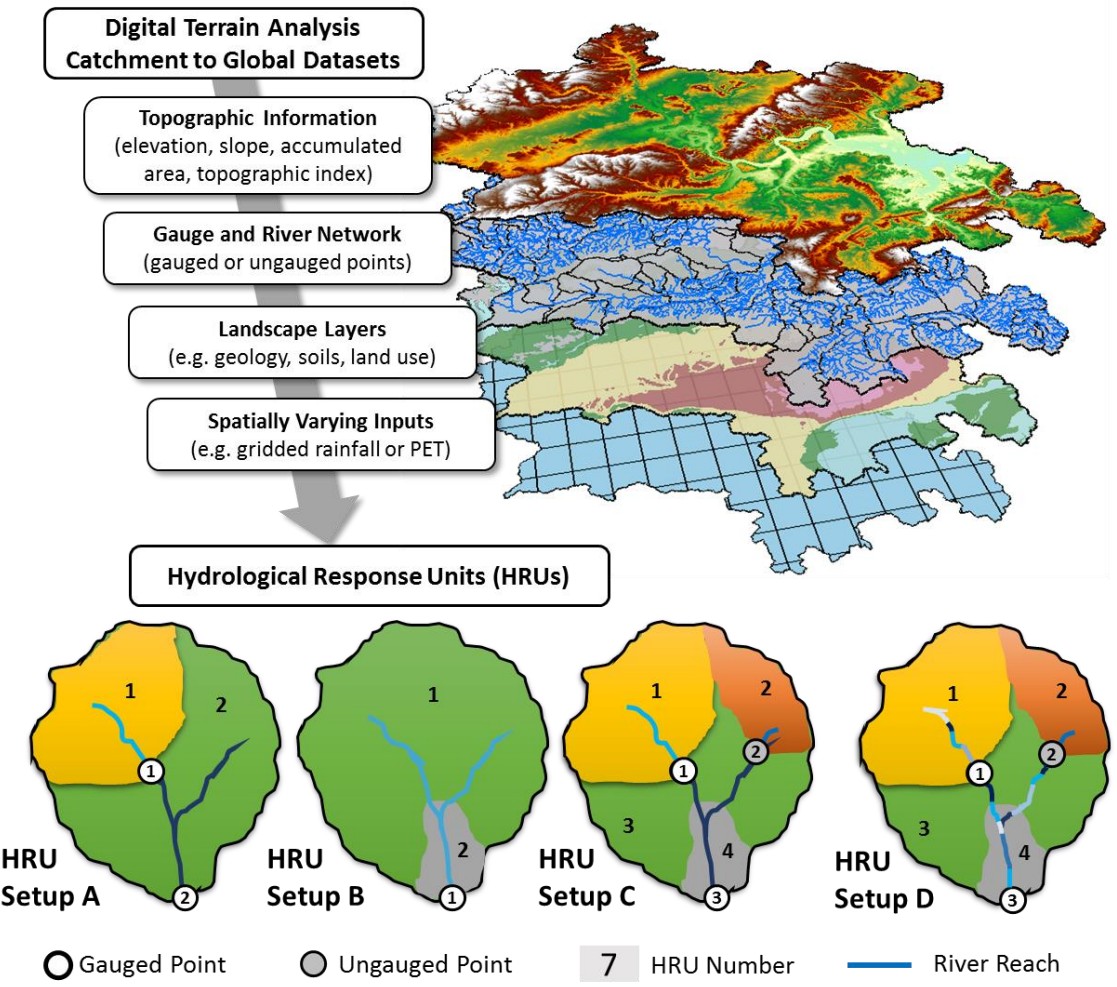

**Figure 1.** Digital Terrain Analysis and simplified examples of using classification layers to discretise a hypothetical catchment into Hydrological Response Units, from a) the gauge network, b) landscape layer with a chalk outcrop for HRU 2, c) the gauge network, ungauged flow point and landscape layer and d) same as c with individual river reach lengths specified

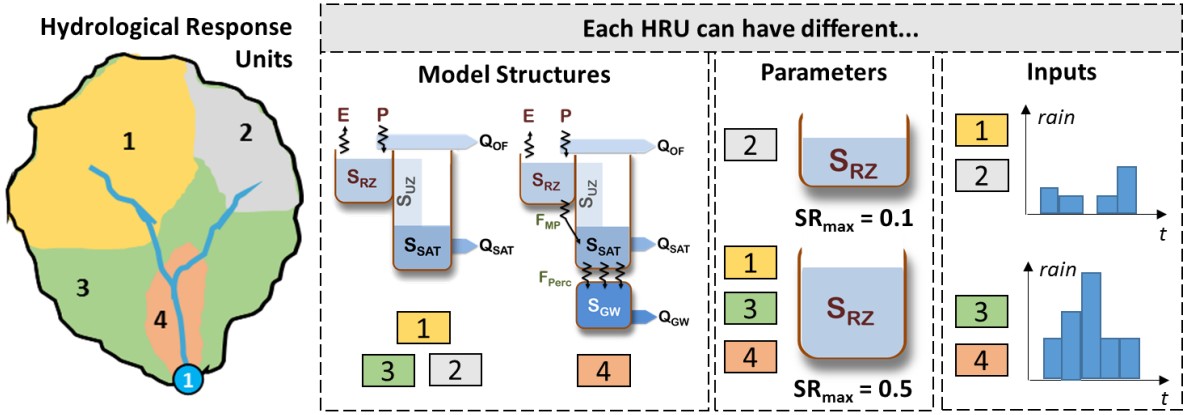

**Figure 2.** DECIPHeR represents spatial heterogeneity in the landscape through hydrological response units (HRUs). Each HRU can have a different model structure, parameters or inputs.

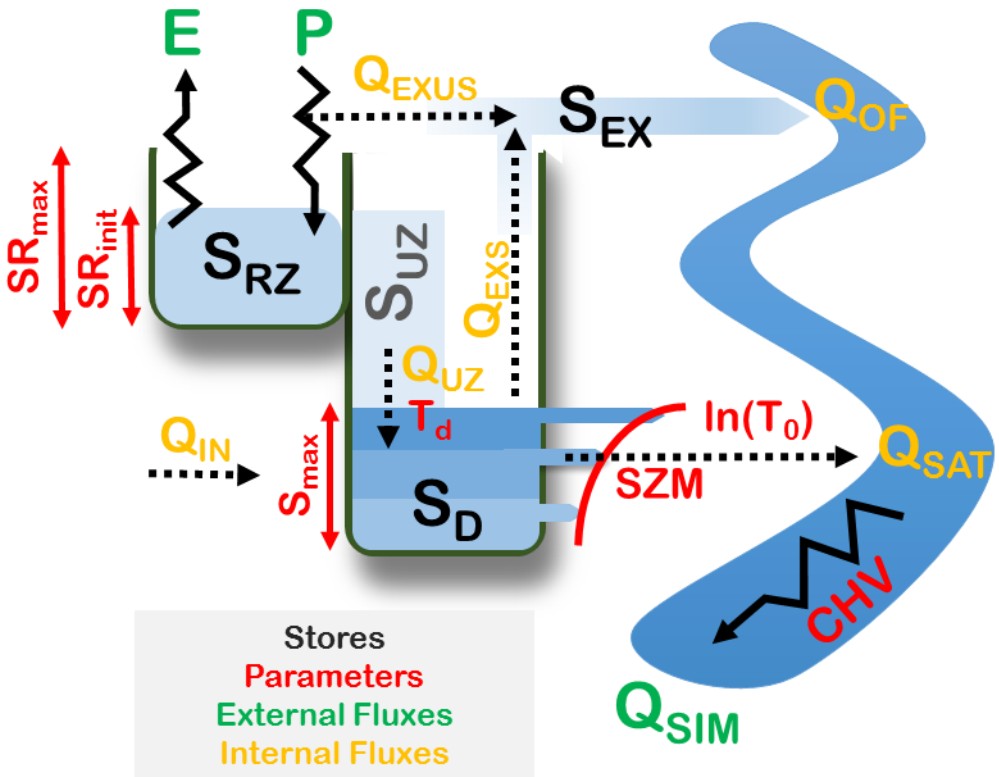

**Figure 3.** Simplified conceptual diagram of the model structure currently implemented in DECIPHeR. All scientific notations are described in Table 1.

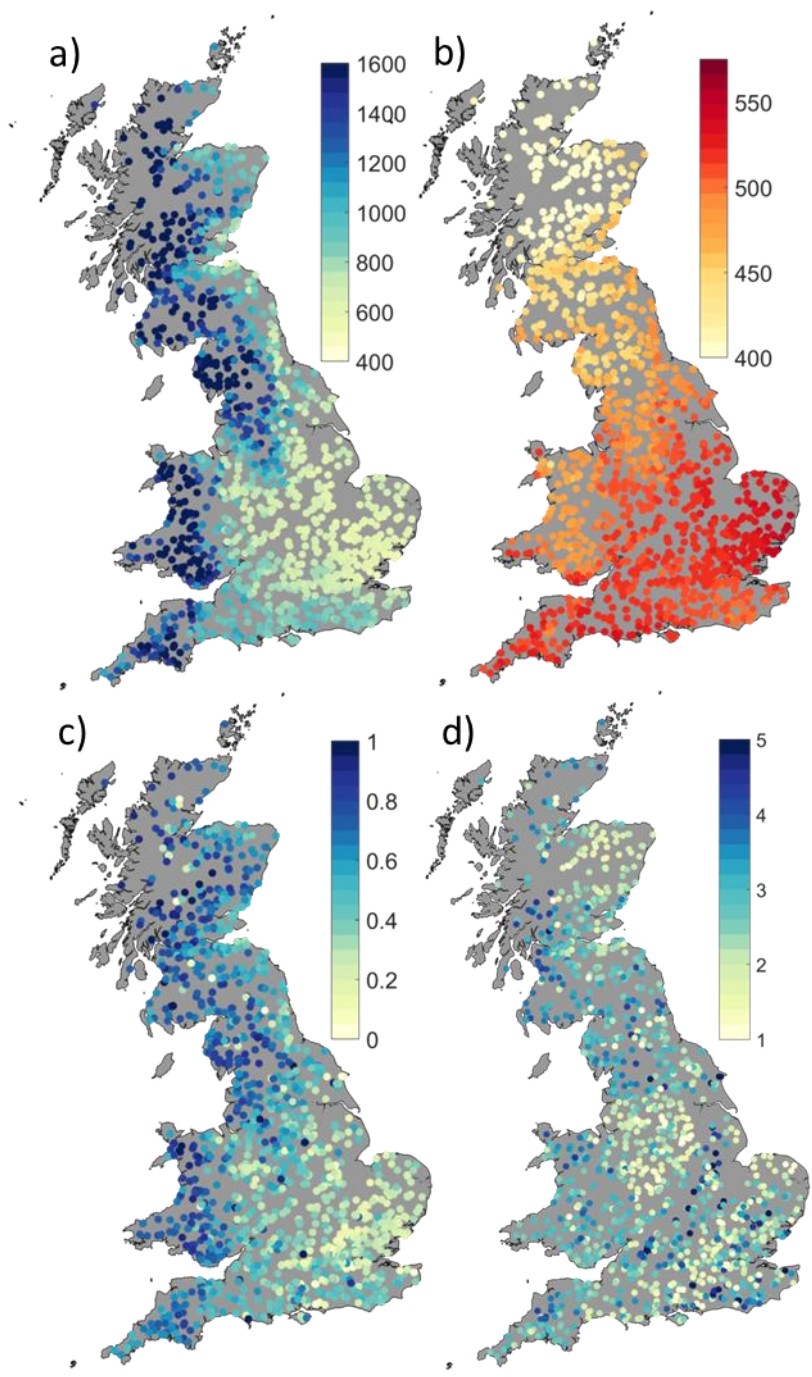

**Figure 4.** Hydro-climatic characteristics of 1366 GB catchments (a) Annual Rainfall (mm/year), (b) Annual potential evapotranspiration (mm/year) (c) Runoff Coefficient (-), d) Slope of the Flow Duration Curve between the 30th and 70th percentiles (-). Min/max values on colorbars have been chosen to show clear differences between catchments.

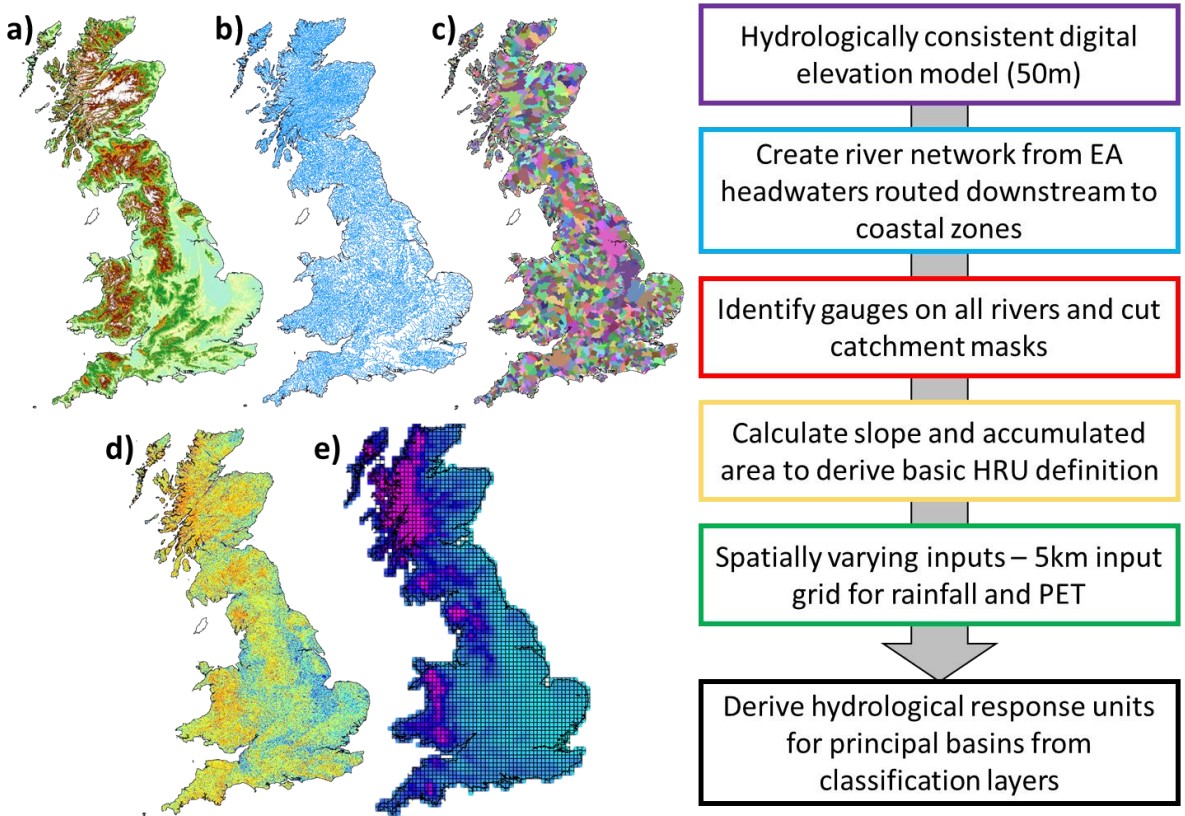

**Figure 5.** Inputs and Outputs of Digital Terrain Analyses for GB a) 50m Hydrologically Consistent Digital Elevation Model, b) DECIPHeR River Network, c) Nested Catchment Mask, d) Topographic Index, e) 5km input grid

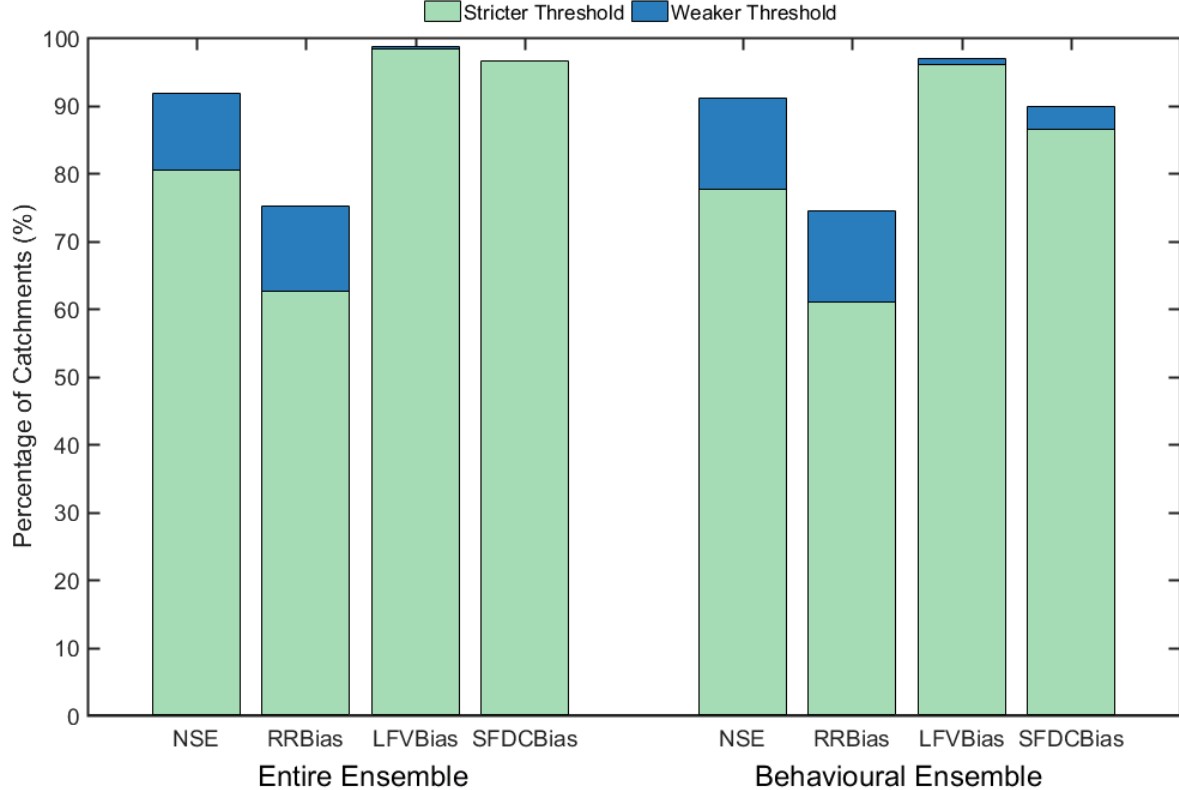

**Figure 6.** Percentage of catchments for each metric that meet the weaker and stricter performance thresholds for the entire ensemble of 10000 model simulations and from the top 1% behavioural ensemble of 100 model simulations generated from the combined ranking of the four metrics.

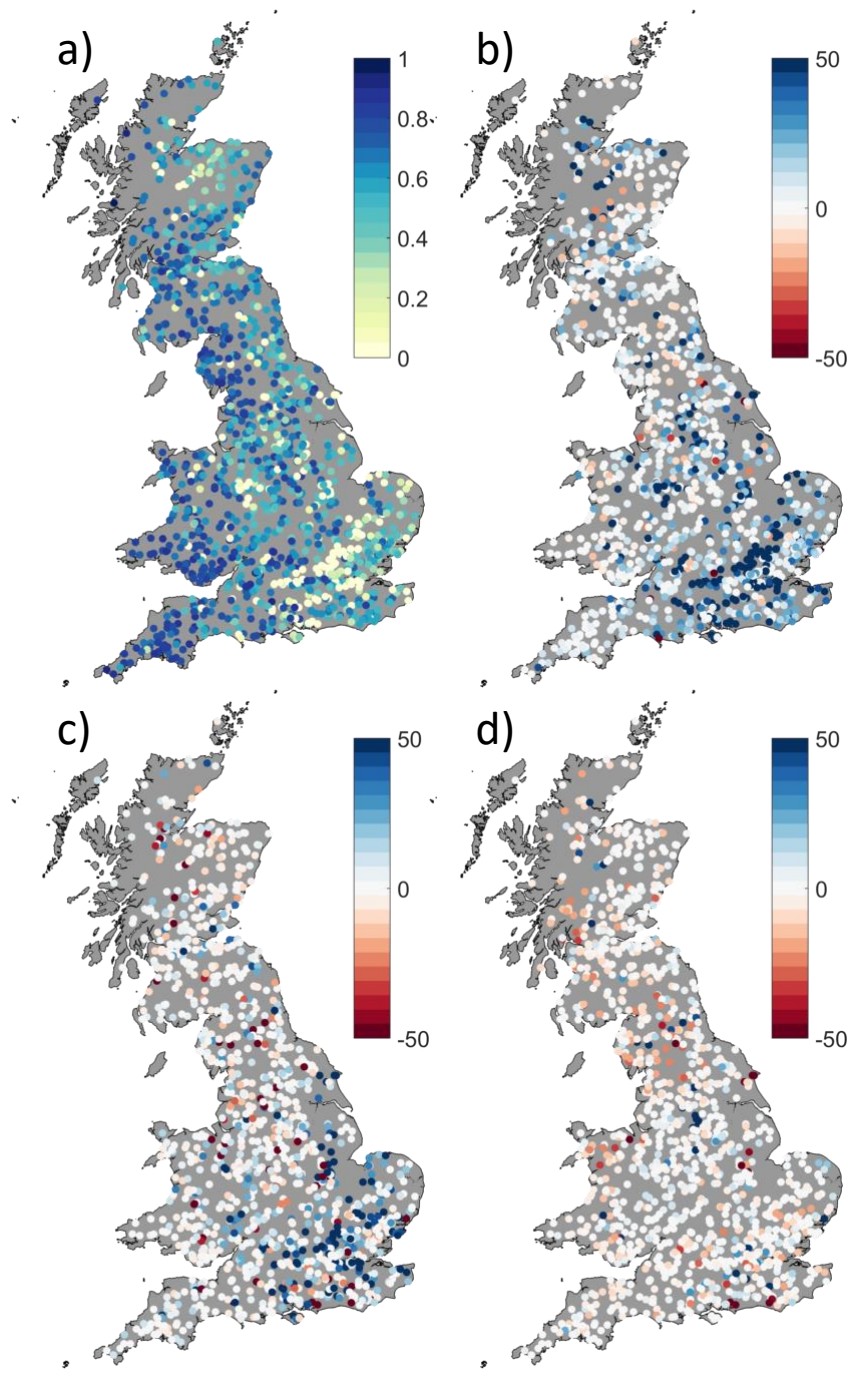

**Figure 7.** Model performance for the best simulation (as defined by the combined rank across all four metrics) for each evaluation metric a) NSE (-), b) Bias in Runoff Ratio (%), c) Bias in Low Flow Volume (%), and d) Bias in Slope of the Flow Duration Curve between the 30$^{th}$ and 70$^{th}$ percentil

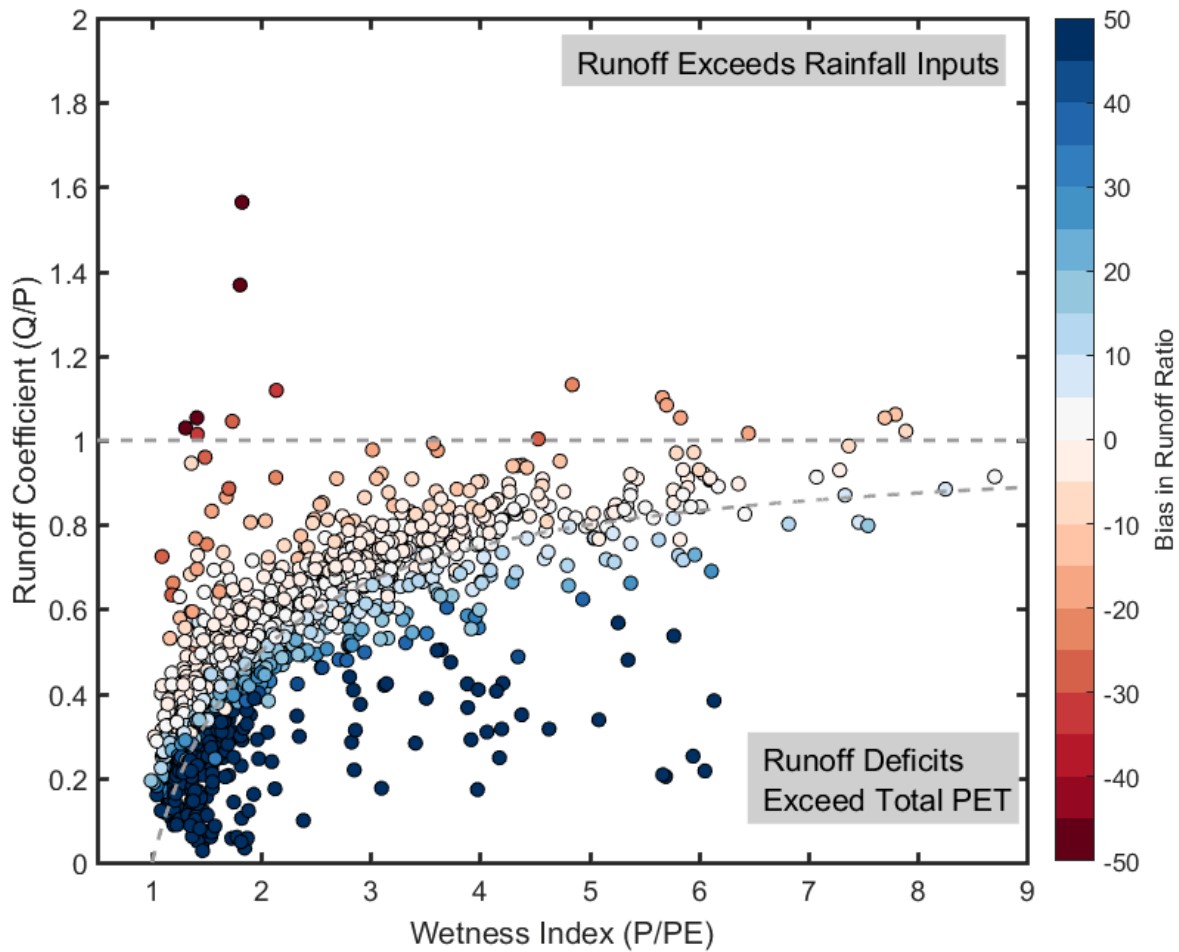

5    **Figure 8.** Scatter plot of wetness index (mean annual precipitation divided by mean annual potential evapotranspiration), runoff coefficient (mean annual discharge divided by mean annual precipitation) and bias in runoff ratio for each GB catchment evaluated in this study.  Any points above the horizontal dotted line are where runoff exceeds total rainfall inputs in a catchment and any points below the curved line are where runoff deficits exceed total potential evapotranspiration in a
10    catchment.

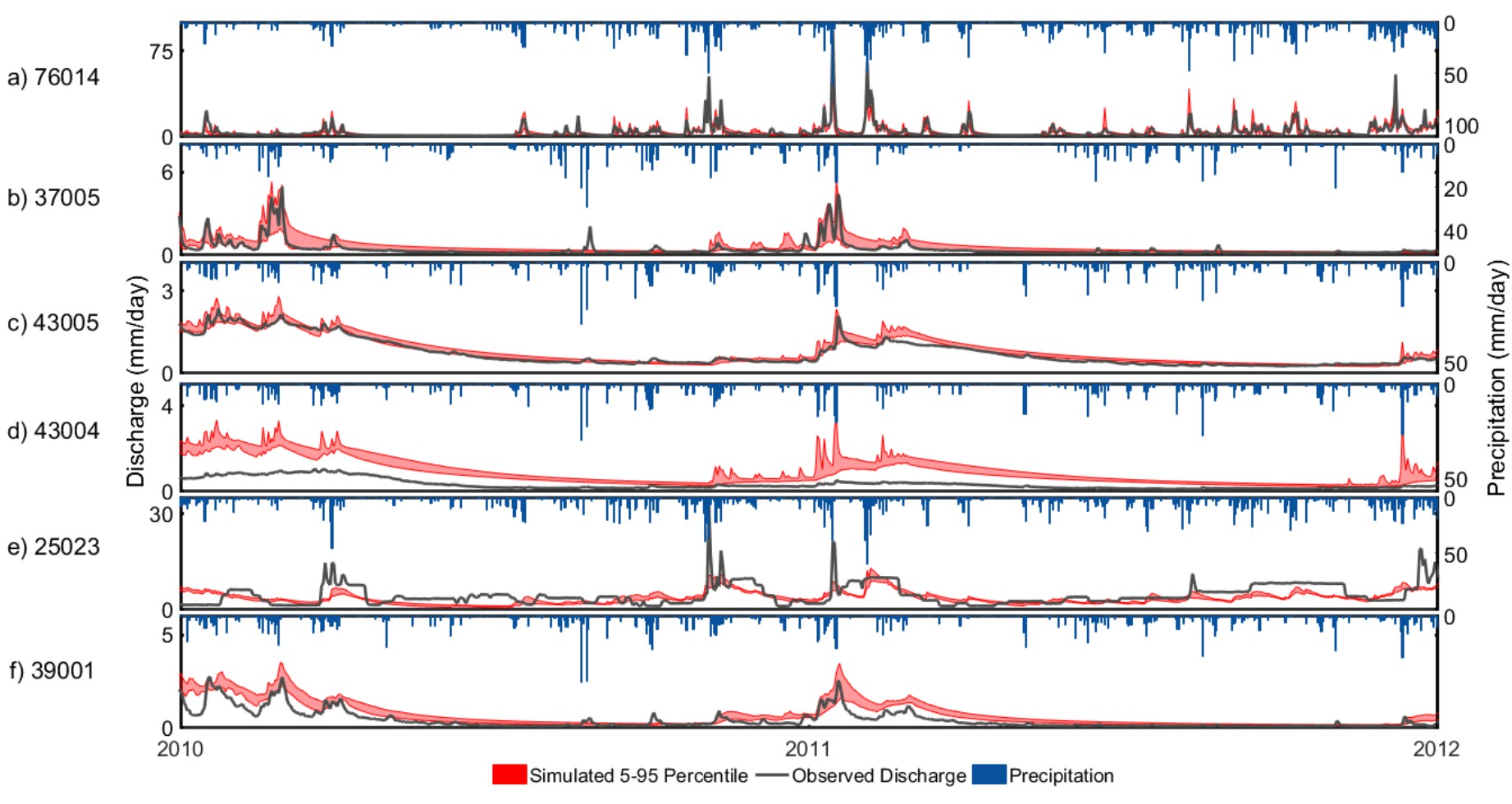

**Figure 9**. Observed discharge and uncertainty bounds for the behavioural simulations ($5^{th}$ and $95^{th}$ percentile of the likelihood-weighted simulated discharge) for six catchments with different characteristics (shown in Table 5). The plots show a two year period (2010-2012) from the 55 year time series simulated.