# Peer review of "DECIPHeR v1: Dynamic fluxEs and ConnectIvity for Predictions of HydRology"

_Geoscientific Model Development, 2018_

## Referee Comment (RC1) · Anonymous Referee #1 · 22 Oct 2018

This paper describes the development of the Dynamic fluxEs and Connectivity for Predictiosn of HydRology (DECIPHeR) framework for simulation of hydrology (especially river flow) at catchment to continental scales. The model is tested across the Great Britain at 1,366 gauges in the current study but the authors intend to expand the model domain and suggest that it can be applied at the continental scales. The framework appears to be efficient computationally but there are a number of issues that authors need to address before the manuscript can be considered for publication. I provide my specific comments below.

(1) The authors should revise the introduction to clearly highlight the motivation behind and the need for such a framework in relation to numerous other ongoing model development efforts. For example, how does the proposed study advance hydrolog-

[Figure]

ical modeling compared to the model presented by Chaney et al. (2016)? Further, there are a number of large-scale models that have the capability to simulate far more number of processes (e.g., groundwater dynamics, pumping, flood dynamics, human impacts) than those presented in the current framework (for example: Hanasaki et al. 2008; Ozdogan et al. 2010; Pokhrel et al. 2015; Wada et al. 2014). Certainly these models are intended for global/regional applications but there have been ongoing efforts to increase the spatial resolution (i.e., hyper-resolution models) for application of these models at smaller scales. Extensive review of these models is available in recent literature (Nazemi and Wheater 2015; Pokhrel et al. 2016; Wada et al. 2017). I suggest that the authors thoroughly revise the introduction including a discussion on these past/ongoing efforts. Note that most of these models use TOPMODEL to simulate some of the surface/sub-surface hydrologic processes.

(2) Since the framework is currently designed to primarily simulate river flow, it is also important to note studies on streamflow/flood simulations at local to continental scales (Bates et al. 2010; Miguez-Macho and Fan 2012; Yamazaki et al. 2011; Zhao et al. 2017). What is the rationale for having the new framework?

(3) The above two issues are important because the authors' intent is to provide a framework for large-scale application.

(4) P4, L16-40: Why did the authors use HRUs instead of doing a fully-distributed model? Is it just the run time minimization? Is there a compromise in terms of adding new features such as groundwater flows and human water use? Again, I suggest adding a note on how this framework advances our capability to simulate the hydrology in comparison to numerous existing framework (see comments above)?

(5) P5, L15: "must contain no sinks": What if there are real inland sinks? There are too many across continents.

(6) Section 2.2.3: What is the routing scheme used? I find some description later in another section. Please consolidate the text and provide more details.

(7) P7, L24: "potential evapotranspiration": first, this term is used here and then abbreviated several times later. Second, why is PET required for rainfall-runoff modeling? Is it to calculate the actual ET? If yes, where is such description provided?

(8) P7, L40: why and how was the 1mm/day set?

(9) P7, L43: what are the "internal states"? Some examples should be provided.

(10) P7, L45: How are runoff generation, infiltration, and soil moisture movement modeled? Are they done in the same manner as in the original TOPMODEL?

(11) P8, L15: What does the "multiple different" refer to?

(12) P8, L24: How is SRmax determined?

(13) P9, L6: "kinematic wave" formulation: is this sufficient when applying the model over large continents where backwater flow and other river-flood dynamics are important (see: Bates et al. 2010; Miguez-Macho and Fan 2012; Yamazaki et al. 2011; Zhao et al. 2017).

(14) P9, L42: "evapotranspiration losses are highest . . ..": The figure shows PET, not the actual ET, and I believe high PET doesn't necessarily mean high ET (in water limited regimes). I think this argument is not supported unless the actual ET is shown. Could the authors clarify this?

(15) P10, L32-L42: Is the river network map described consistent with the topography data described in the previous paragraph? Isn't it necessary to generate a river network map from the DEM used in the model?

(16) Section 3.3.1: Are the precip data used here same as those shown in Fig. 3.1?

(17) Section 3.3.2: What are the calibration and validation periods?

(18) P11, L14-25: what is the use of PET here? In fact, it was not clear to me on what the forcing variables are. Typical hydrological models use Precip, Temp, Radiation,

Humidity, Wind etc. If such variables are used, is the PET consistent with those forcing variables?

(19) Section 3.4.3 (P13, L33): The authors should present the actual time streamflow time series. Since this is the only the variable simulated/discussed, I was surprised that authors are not showing the time series plots. I suggest selecting certain representative gauging stations with varying catchment area and those located in different climatic regions for such analysis (it could be a 20 stations for example).

(20) Then, I also suggest showing the annual mean flow (rate or volume) as a scatter diagram for all gauging stations. Evaluation of high (Q5) and low (e.g., Q95) can also be presented similarly. Overall, the validation provided in the current version is not satisfactory/sufficient.

(21) P14, L23: "time series": where is this shown?

(22) P14, L39-45: The authors could discuss the appropriateness of different performance measures by referring some recent studies that have used a wide range of such performance measures (Veldkamp et al. 2018; Zaherpour et al. 2018). This comment is relevant to P12, L5-15 as well.

(23) P15, L23: "groundwater dynamics and human influences": Is the HRU-based representation a suitable choice for the representation of these missing factors? Would a fully distributed be required? Please also see a related comment earlier.

(24) Finally, the authors should provide caveats in the current framework and the challenges in upscaling the framework to continental and possibly to global scales. The discussion regarding advancements compared to the existing models/ongoing efforts (e.g., the National Water Model) also becomes relevant here. A note on the use on the use HRUs, and not distributed grids, should also be made.

Minor/editorial issues: (25) P2, L2: impact on "what"?

(26) P2, L8: some refs contain first names/initials

(27) P11, L8: PET is abbreviated here but already used before.

(28) P12, L32: the catchment details are redundant with the information in Section 3.

References: Bates, P. D., M. S. Horritt, and T. J. Fewtrell, 2010: A simple inertial formulation of the shallow water equations for efficient two-dimensional flood inundation modelling. Journal of Hydrology, 387, 33-45.

Chaney, N. W., P. Metcalfe, and E. F. Wood, 2016: HydroBlocks: a field‐scale resolving land surface model for application over continental extents. Hydrol. Process., 30, 3543-3559.

Hanasaki, N., S. Kanae, T. Oki, K. Masuda, K. Motoya, N. Shirakawa, Y. Shen, and K. Tanaka, 2008: An integrated model for the assessment of global water resources – Part 1: Model description and input meteorological forcing. Hydrol. Earth Syst. Sci., 12, 1007-1025.

Miguez-Macho, G., and Y. Fan, 2012: The role of groundwater in the Amazon water cycle: 1. Influence on seasonal streamflow, flooding and wetlands. J. Geophys. Res. Atmos., 117, D15113.

Nazemi, A., and H. S. Wheater, 2015: On inclusion of water resource management in Earth system models – Part 1: Problem definition and representation of water demand. Hydrol. Earth Syst. Sci., 19, 33-61.

Ozdogan, M., M. Rodell, H. K. Beaudoing, and D. L. Toll, 2010: Simulating the Effects of Irrigation over the United States in a Land Surface Model Based on Satellite-Derived Agricultural Data. Journal of Hydrometeorology, 11, 171-184.

Pokhrel, Y., N. Hanasaki, Y. Wada, and H. Kim, 2016: Recent progresses in incorporating human land–water management into global land surface models toward their integration into Earth system models. WIREs Water, 3, 548-574.

Pokhrel, Y., S. Koirala, P. J. F. Yeh, N. Hanasaki, L. Longuevergne, S. Kanae, and T.

Oki, 2015: Incorporation of groundwater pumping in a global Land Surface Model with the representation of human impacts. Water Resources Research, 51, 78-96.

Veldkamp, T. I. E., F. Zhao, P. J. Ward, H. de Moel, J. C. Aerts, H. M. Schmied, F. T. Portmann, Y. Masaki, Y. Pokhrel, and X. Liu, 2018: Human impact parameterizations in global hydrological models improve estimates of monthly discharges and hydrological extremes: a multi-model validation study. Environmental Research Letters, 13, 055008.

Wada, Y., D. Wisser, and M. F. P. Bierkens, 2014: Global modeling of withdrawal, allocation and consumptive use of surface water and groundwater resources. Earth Syst. Dynam., 5, 15-40. Wada, Y., M. F. P. Bierkens, A. de Roo, P. A. Dirmeyer, J. S. Famiglietti, N. Hanasaki, M. Konar, J. Liu, H. Müller Schmied, T. Oki, Y. Pokhrel, M. Sivapalan, T. J. Troy, A. I. J. M. van Dijk, T. van Emmerik, M. H. J. Van Huijgevoort, H. A. J. Van Lanen, C. J. Vörösmarty, N. Wanders, and H. Wheater, 2017: Human–water interface in hydrological modelling: current status and future directions. Hydrol. Earth Syst. Sci., 21, 4169-4193.

Yamazaki, D., S. Kanae, H. Kim, and T. Oki, 2011: A physically based description of floodplain inundation dynamics in a global river routing model. Water Resources Research, 47, W04501. Zaherpour, J., S. N. Gosling, N. Mount, H. M. Schmied, T. I. E. Veldkamp, R. Dankers, S. Eisner, D. Gerten, L. Gudmundsson, and I. Haddeland, 2018: Worldwide evaluation of mean and extreme runoff from six global-scale hydrological models that account for human impacts. Environmental Research Letters.

Zhao, F., T. I. Veldkamp, K. Frieler, J. Schewe, S. Ostberg, S. Willner, B. Schauberger, S. N. Gosling, H. M. Schmied, F. T. Portmann, G. Leng, M. Huang, X. Liu, Q. Tang, N. Hanasaki, H. Biemans, D. gerten, Y. Satoh, Y. Pokhrel, T. stacke, P. ciais, J. Chang, A. Ducharne, M. Guimberteau, Y. Wada, H. kim, and D. Yamazaki, 2017: The critical role of the routing scheme in simulating peak river discharge in global hydrological models. Environmental Research Letters, 12, 075003.

---

## Referee Comment (RC2) · Anonymous Referee #2 · 5 Nov 2018

General comments

The authors extended the concept and code of Dynamic TOPMODEL and developed an improved model termed DECIPHeR v1. They applied it to the entire Great Britain by calibrating and validating at 1366 gauges, and claimed that the performance was satisfactory. As a hydrological modeler who has developed open source code and applied it to an extensive study domain, I fully acknowledge the considerable efforts the authors made. The paper is overall readable for most parts but seems lacking some important statements particularly on the novelty and originality. The key characteristics and strengths of DECIPHeR should be clearly stated in comparison with existing catchment and global hydrological models (the current form of paper only compares DECIPHeR with the original Dynamic TOPMODEL). Also the value and significance

of the model application to the entire Great Britain should be further discussed (the current form displays the performance scores without referring any earlier efforts).

Specific comments

Page 1 Line 15: "a new flexible model framework": Make this part more specific. What is a flexible model framework (or what is an inflexible model)? Also, add the key strengths and characteristics of DECIPHeR compared to existing hydrological models.

Page 1 Line 18: "modified to represent different levels of heterogeneity, connectivity and hydrological processes as needed": Make this part more specific. All models can be "modified to represent" these in some extent. Add more concrete words in what sense DECIPHer is more adaptable compared with other models.

Page 2 Line 30 "the underlying model structures do not have the flexibility to represent different levels of complexity in different landscapes": Quite unclear. Since this part is crucially important to identify the research needs/questions, discuss concretely what have been already achieved and what are still lacking by earlier models.

Page 2 Line 42: "This is despite significant development of various modeling tools . . .": Again quite unclear. What have been already achieved and what are still lacking by earlier models?

Page 3 Line 36 "builds on the code and key concepts of Dynamic TOPMODEL.": This sounds that DECIPHeR is an upgrade of Dynamic TOPMODEL. If this is the case, it is more readable to introduce the concept and formulations of Dynamic TOPMODEL first, then show the new functions and characteristics of DECIPHeR. Actually, the present form is hard to know what are same or different between two models.

Page 4 Line 18 "To realise this, DECIPHeR uses hydrological response units (HRUs)": It is hard to know whether the HRV concept has been already included in Dynamic TOPMODEL or not. I was confused similarly by many parts in this section. As mentioned earlier, please make it clear what are same or different between two models

more clearly.

Page 6 Line 9 "In DECIPHeR, they provide the basis for river routing . . .". Ibid.

Page 8 Line 12 "2.3.5 Model Structure": Unfortunately, I could hardly understand the model structure. Please describe all the equations for the terms in Figure 3 and the parameters in Table 1. At least describe where such full description of equations is available.

Page 9 Line 9 "The parameter, SZM, sets..": This paragraph is particularly hard to follow. Please show the key equations how these parameters work.

Page 12 Line 44 "3.4.2 Overall model performance" and Figure 6: I am wondering why the parameters are so insensitive to the results (i.e. it is surprising that 90% of parameter sets yield NSE >0). I am also puzzled why the entire ensemble outperforms the behavioral ensemble (top 1% performance, if I understood correctly). Please elaborate these points.

Page 14 Line 21 "We calculated four evaluation metrics for 10,000 model simulations for 1366 GB gauges. . .": Is this the first study to apply a hydrological to the entire Great Britain? If it is the case, clearly state so. If it is not, clearly refer the earlier efforts and compare the performance of them with this study.

Technical comments

Page 7 Line 27 "a parameter file specifying set parameter bounds for Monte-Carlo-sampling": Is "set" needed?

Page 7 Line 42 Q_SAT: I guess this term first appears. Define what this term is.

Page 12 Line 37: "13,600,600" reads 13,660,000.

Page 13 Line 4"The vast majority of gauges (90% of the whole ensemble)": 90% of the gauges or 90% of the ensemble (i.e. 9000 simulations)?

Page 14 Line 27: "is" reads in.

Page 34 figure 6: The caption says "weaker and stricter" while the figure says "upper and lower".

---

## Author Response (AR1)

**GMD Submission by Coxon et al**
**DECIPHeR v1: Dynamic fluxEs and ConnectIvity for Predictions of HydRology**

**General Response**

We thank the reviewers for taking the time to review the paper and their comments, which have greatly helped to improve this manuscript and the quality and clarity of the research.

The main comments from the reviewers focused on (1) better defining the novelty and originality of the proposed modelling framework, (2) clarity of the model structure and equations and (3) the model evaluation.

In response to these reviewer comments, we have substantially re-written sections of the abstract, introduction and key concepts to highlight the unique features of the modelling framework and how it compares to other modelling frameworks in hydrology. We have produced flow timeseries plots for a new figure in the paper that demonstrates the ability of the model to reproduce the observed flow timeseries for six catchments.

Due to the requested extra time the GMD editor kindly agreed to, we have also been able to implement a new, more computationally efficient and stable, analytical solution to the subsurface flow equations and have detailed this solution in a new appendix. The result of this has a) increased the novelty of the model because the solutions are a departure from those implemented in Dynamic TOPMODEL (Beven and Freer, 2001), b) resulted in improvements in the national results overall, thus all figures/results now reflect these new equations, c) sped up simulation times as the solution is no longer iterative and d) addressed the comments of Reviewer 2 who requested more detail about the implementation of our flux equations.

Detailed responses to all comments are provided below. Author responses are in **bold** and any modifications to the manuscript are in *italic* below each of the reviewer's comments. The reviewer comments are inserted as comments next to the relevant tracked changes in the main document.

Gemma Coxon, March 2019

**Reviewer #1**

This paper describes the development of the Dynamic fluxEs and Connectivity for Predictiosn of HydRology (DECIPHeR) framework for simulation of hydrology (especially river flow) at catchment to continental scales. The model is tested across the Great Britain at 1,366 gauges in the current study but the authors intend to expand the model domain and suggest that it can be applied at the continental scales. The framework appears to be efficient computationally but there are a number of issues that authors need to address before the manuscript can be considered for publication. I provide my specific comments below.

**We thank Reviewer #1 for taking the time to review the paper. We appreciate their comments and provide our responses below.**

(1) The authors should revise the introduction to clearly highlight the motivation behind and the need for such a framework in relation to numerous other ongoing model development efforts. For example, how does the proposed study advance hydrological modeling compared to the model presented by Chaney et al. (2016)? Further, there are a number of large-scale models that have the capability to simulate far more number of processes (e.g., groundwater dynamics, pumping, flood dynamics, human impacts) than those presented in the current framework (for example: Hanasaki et al. 2008; Ozdogan et al. 2010; Pokhrel et al. 2015; Wada et al. 2014). Certainly these models are intended for global/regional applications but there have been ongoing efforts to increase the spatial resolution (i.e., hyper-resolution models) for application of these models at smaller scales. Extensive review of these models is available in recent literature (Nazemi and Wheater 2015; Pokhrel et al. 2016; Wada et al. 2017). I suggest that the authors thoroughly revise the introduction including a discussion on these past/ongoing efforts. Note that most of these models use TOPMODEL to simulate some of the surface/sub-surface hydrologic processes.

**We agree that we needed to make this clearer and have revised the abstract and introduction to clearly highlight the motivation behind the framework and it's unique features in response to comments from Reviewer 1 and 2.**

(2) Since the framework is currently designed to primarily simulate river flow, it is also important to note studies on streamflow/flood simulations at local to continental scales (Bates et al. 2010; Miguez-Macho and Fan 2012; Yamazaki et al. 2011; Zhao et al. 2017). What is the rationale for having the new framework?

**We do not believe the studies suggested by the reviewer are relevant to this study. Bates et al (2010) presents a new set of equations for floodplain inundation, Miguez-Macho and Fan (2012) are focused on the role of groundwater and Yamazaki et al (2011) and Zhao et al (2017) are primarily investigating flow routing schemes. While these are interesting studies, they are not focused on modelling frameworks that simulate the key hydrological processes (e.g. infiltration, runoff generation, subsurface flows etc.) at catchment-based scales in the generation of river flow and thus are fundamentally different to the modelling framework presented here.**

(3) The above two issues are important because the authors' intent is to provide a framework for large-scale application.

**We agree that this is an important point and address this comment in the two responses above.**

(4) P4, L16-40: Why did the authors use HRUs instead of doing a fully-distributed model? Is it just the run time minimization? Is there a compromise in terms of adding new features such as groundwater flows and human water use? Again, I suggest adding a note on how this framework advances our capability to simulate the hydrology in comparison to numerous existing framework (see comments above)?

**One clear benefit of using HRUs is minimising the run times of the model. However, the key benefit is the flexibility it gives you to modify the spatial complexity/scale of how spatial variability and hydrologic connectivity are represented. This flexibility means you can (1) run the model as a fully distributed, semi-distributed or lumped model, (2) have more/less spatial/process complexity where needed in the landscape and (3) represent point scale features in the landscape whilst still maintaining modelling efficiencies elsewhere. These features are hugely beneficial to having a pre-defined fully distributed model which cannot handle such occurrences. We have modified section 2.1 to clarify this point.**

**There isn't any compromise in terms of adding new features as each HRU is treated as a separate store in the model which can have different process conceptualisations and parameterisations. This means that more process complexity can be incorporated where needed to better suit local conditions e.g. to account for 'point-source' human influences or more complex hydrological processes such as surface-groundwater exchanges.**

(5) P5, L15: "must contain no sinks": What if there are real inland sinks? There are too many across continents.

**Sink filling is very common in digital terrain analyses when generating river and catchment layers (for example the SRTM DEM used for HydroSheds undergoes a sink filling process before it can be used to derive catchment basins). We agree that this will mean any real inland sinks in the digital elevation model will be filled. Currently the modelling framework is unable to account for these features (such as lakes), however, this is a feature we will be looking to include soon.**

(6) Section 2.2.3: What is the routing scheme used? I find some description later in another section. Please consolidate the text and provide more details.

**In Section 2.2.3 (now Section 2.2.2) we are describing the river routing data that is generated by the digital terrain analysis. These data (such as the river network connectivity and routing tables) provide the information for several different routing schemes. The river routing scheme is then described fully in the model structure (Section 2.3.4) as this is the current routing scheme implemented in the model.**

**We have modified section 2.2.2 to guide the reader better:**

*"From the river network and gauge locations, the river network connectivity is derived with each river section labelled with a unique river ID and a suite of routing tables so that each ID knows it's downstream connections and to allow multiple routing schemes to be configured (see section 2.3.4 for a description of the current routing scheme implemented in the modelling framework)."*

(7) P7, L24: "potential evapotranspiration": first, this term is used here and then abbreviated several times later. Second, why is PET required for rainfall-runoff modeling? Is it to calculate the actual ET? If yes, where is such description provided?

**We have removed all abbreviations of potential evapotranspiration in the document. Potential evapotranspiration is a common input for hydrological models and is used to calculate actual ET. The description of how this is calculated is given by Equation 2 in Section 2.3.4, but the model could also include other conceptualisations depending on the users requirements.**

(8) P7, L40: why and how was the 1mm/day set?

**The model needs a starting flow to initialise the storage deficits. Typically we take this from an observed flow time series but in some cases, particularly for ungauged flow points, there may be no flow time series available. In this case we define a starting flow of 1mm/day as a representative starting flow for most catchments. The choice of this initial starting flow only affects model flows during the initialisation period and has no effect once the flows are fully initialised. The model is always started with a 'spin up' period as would be normal standard practice.**

(9) P7, L43: what are the "internal states"? Some examples should be provided.

**We have modified this to** *"model stores and fluxes"* **to better clarify this point. These are described in full in Section 2.3.4.**

(10) P7, L45: How are runoff generation, infiltration, and soil moisture movement modeled? Are they done in the same manner as in the original TOPMODEL?

**These processes are described fully in Section 2.3.5 focused on the model structure. We have made clearer the differences between Dynamic TOPMODEL and DECIPHeR in Section 2.1 in response to Reviewer 2.**

(11) P8, L15: What does the "multiple different" refer to?

**In this case it refers to the model structures i.e. you can implement many different types of model structure within the model framework. We have removed the word 'different' to better clarify this point.**

(12) P8, L24: How is SRmax determined?

**SRmax is a parameter within the model that determines the soil root zone. The user can either set this to a default value or it can be sampled from parameter bounds as explained in section 2.3.3.**

(13) P9, L6: "kinematic wave" formulation: is this sufficient when applying the model over large continents where backwater flow and other river-flood dynamics are important (see: Bates et al. 2010; Miguez-Macho and Fan 2012; Yamazaki et al. 2011; Zhao et al. 2017).

**We believe the reviewer has mis-interpreted the routing used in the model. Channel river flow routing in the model is modelled using a set of time delay histogram. We agree that using a set of time delay histograms may not be appropriate where backwater flow and other river-flood dynamics may be important. However, we would like to stress that the model is not intended to be a flood inundation model and is not trying to compute full hydrodynamics. Computation times for such models are significantly longer than here (hours rather than minutes for simulations over 30-40 years).**

**The model is flexible to accommodate other flow routing schemes (as discussed in Section 2.2.2) and allow for variability in channel routing at the reach scale to recognise changes in local routing velocities. This will certainly be an area of future research to**

**improve the channel river flow routing. We have also recently coupled the model to LISFLOOD-FP to provide a better representation of river-flood dynamics in regions where this is important.**

(14) P9, L42: "evapotranspiration losses are highest . . ..": The figure shows PET, not the actual ET, and I believe high PET doesn't necessarily mean high ET (in water limited regimes). I think this argument is not supported unless the actual ET is shown. Could the authors clarify this?

**We have modified the text to clarify the data are potential evapotranspiration and not actual evapotranspiration.**

(15) P10, L32-L42: Is the river network map described consistent with the topography data described in the previous paragraph? Isn't it necessary to generate a river network map from the DEM used in the model?

**Yes, we ensure consistency between the river and the DEM by producing the river network used by the model from the DEM during the digital terrain analysis. As described in section 3.2, we extract headwater cells from an external river map (the Ordnance Survey MasterMap Water Network Layer) and then route these cells downstream via the steepest slope so that the DEM and the calculated stream network are consistent for flow accumulations based on surface slope. Consequently, it is generated from the DEM used in the model and thus consistent. We have better clarified this point in Section 3.2 to avoid confusion.**

(16) Section 3.3.1: Are the precip data used here same as those shown in Fig. 3.1?

**We believe the reviewer is referring to Figure 4a here. The data used to derive the hydro-climatic characteristics are the same as the model forcing data described in Section 3.3.1. We have made this point clearer in the text.**

(17) Section 3.3.2: What are the calibration and validation periods?

**As described in Section 3.3.1, daily data of precipitation, potential evapotranspiration and discharge for a 55-year period from 01/01/1961–31/12/2015 were used to run and assess the model. The year 1961 was used as a warm-up period for the model; therefore no model evaluation was quantified in this period and the model was evaluated from 01/01/1962 – 31/12/2015. In this study we don't use a split sample test with a calibration and validation period and instead choose to evaluate the model for the full time series available.**

(18) P11, L14-25: what is the use of PET here? In fact, it was not clear to me on what the forcing variables are. Typical hydrological models use Precip, Temp, Radiation, Humidity, Wind etc. If such variables are used, is the PET consistent with those forcing variables?

**Potential evapotranspiration is required as a forcing time-series for DECIPHeR to calculate actual evapo-transpiration. This is described in Section 2.3.2 which outlines the Data Pre-requisites of the model and Equation 2 in Section 2.3.5 which describes how potential evapotranspiration is used to calculate actual evapotranspiration.**

**We do not fully agree that 'typical' hydrological models use a range of data to construct PET internally. There are many hydrological models that use PET directly. Given there is considerable differences in how to construct PET then we often use more than one method to explore differences.**

**In this study (as described in Section 3.3.1), daily potential evapotranspiration (PET) data were obtained from the CEH Climate hydrology and ecology research support system potential evapotranspiration dataset for Great Britain (CHESS-PE) (Robinson et al., 2016). This dataset consists of 1km$^2$ gridded estimates of daily potential evapotranspiration for Great Britain from 1961 - 2015 calculated using the Penman-Monteith equation and data from the CHESS meteorology dataset (in this case air temperature, specific humidity, downward long- and shortwave radiation and surface air pressure). Consequently, potential evapotranspiration is calculated before being used as an input to DECIPHeR (as is common for many hydrological models).**

(19) Section 3.4.3 (P13, L33): The authors should present the actual time streamflow time series. Since this is the only the variable simulated/discussed, I was surprised that authors are not showing the time series plots. I suggest selecting certain representative gauging stations with varying catchment area and those located in different climatic regions for such analysis (it could be a 20 stations for example).

**Thank you for this suggestion. We have added a new figure (Figure 9) and text (Section 3.4.5) to the manuscript that shows the flow time series results for six gauging stations.**

(20) Then, I also suggest showing the annual mean flow (rate or volume) as a scatter diagram for all gauging stations. Evaluation of high (Q5) and low (e.g., Q95) can also be presented similarly. Overall, the validation provided in the current version is not satisfactory/sufficient.

**The model is evaluated against a large sample of catchments (1,366) for a number of different metrics capturing the annual flow rate (bias in runoff ratio), low flows (bias in low flow volume) and high flows (nash-sutcliffe efficiency). While we appreciate the reviewer's suggestions, we present results that already evaluate the model's ability to capture these aspects of the flow regime (see Figure 7). The main aim of the paper is to provide a description of DECIPHeR and more detailed model evaluation is outside the scope of this paper.**

(21) P14, L23: "time series": where is this shown?

**This is now shown in the new figure 9.**

(22) P14, L39-45: The authors could discuss the appropriateness of different performance measures by referring some recent studies that have used a wide range of such performance measures (Veldkamp et al. 2018; Zaherpour et al. 2018). This comment is relevant to P12, L5-15 as well.

**Thanks for the suggestions. We have added these references in section 4.1.**

(23) P15, L23: "groundwater dynamics and human influences": Is the HRU-based representation a suitable choice for the representation of these missing factors? Would a fully distributed be required? Please also see a related comment earlier.

**Please see response to comment 4 above.**

(24) Finally, the authors should provide caveats in the current framework and the challenges in upscaling the framework to continental and possibly to global scales. The discussion regarding advancements compared to the existing models/ongoing efforts (e.g., the National Water Model) also becomes relevant here. A note on the use on the use HRUs, and not distributed grids, should also be made.

Section 4.2 and 4.3 discuss the limitations of the modelling framework as applied in this study and the areas for future research. These limitations are very relevant to applying the framework across continental scales and we have made this clearer in the discussion.

**Minor/editorial issues:**

(25) P2, L2: impact on "what"?

**We have modified sentence**

(26) P2, L8: some refs contain first names/initials

**We have removed the first name and initial from this reference**

(27) P11, L8: PET is abbreviated here but already used before.

**We have modified to include only the abbreviation**

(28) P12, L32: the catchment details are redundant with the information in Section 3.

**We disagree and believe these catchment details are essential information.**

**Reviewer #2**

General comments

The authors extended the concept and code of Dynamic TOPMODEL and developed an improved model termed DECIPHeR v1. They applied it to the entire Great Britain by calibrating and validating at 1366 gauges, and claimed that the performance was satisfactory. As a hydrological modeler who has developed open source code and applied it to an extensive study domain, I fully acknowledge the considerable efforts the authors made. The paper is overall readable for most parts but seems lacking some important statements particularly on the novelty and originality. The key characteristics and strengths of DECIPHeR should be clearly stated in comparison with existing catchment and global hydrological models (the current form of paper only compares DECIPHeR with the original Dynamic TOPMODEL). Also the value and significance of the model application to the entire Great Britain should be further discussed (the current form displays the performance scores without referring any earlier efforts).

**We thank Reviewer #2 for taking the time to review the paper. We appreciate their comments and provide responses below.**

Specific comments

Page 1 Line 15: "a new flexible model framework": Make this part more specific. What is a flexible model framework (or what is an inflexible model)? Also, add the key strengths and characteristics of DECIPHeR compared to existing hydrological models.

**We have modified the abstract to be more specific on the flexibility of the model framework (see response to comment below).**

**The key strengths and characteristics of DECIPHeR compared to existing hydrological models is now better discussed in the introduction in response to the comments of Reviewer #1 – we feel this discussion is more appropriate in the introduction rather than the abstract.**

Page 1 Line 18: "modified to represent different levels of heterogeneity, connectivity and hydrological processes as needed": Make this part more specific. All models can be "modified to represent" these in some extent. Add more concrete words in what sense DECIPHer is more adaptable compared with other models.

**We have modified the abstract to be more specific on the flexibility/adaptability of the model framework.**

*"This paper presents DECIPHeR (Dynamic fluxEs and ConnectIvity for Predictions of HydRology); a new model framework that simulates and predicts hydrologic flows from spatial scales of small headwater catchments to entire continents. DECIPHeR can be adapted to specific hydrologic settings and to different levels of data availability. It is a flexible model framework which has the capability to (1) change its representation of spatial variability and hydrologic connectivity by implementing hydrological response units in any configuration, and (2) test different hypotheses of catchment behaviour by altering the model equations and parameters in different parts of the landscape."*

Page 2 Line 30 "the underlying model structures do not have the flexibility to represent different levels of complexity in different landscapes": Quite unclear. Since this part is

crucially important to identify the research needs/questions, discuss concretely what have been already achieved and what are still lacking by earlier models.

**We agree this could be made clearer and have removed this sentence. We have significantly rewritten the introduction in response to comments from both reviewers to clarify the novelty of DECIPHeR and it's differences to other modelling frameworks.**

Page 2 Line 42: "This is despite significant development of various modeling tools . . .": Again quite unclear. What have been already achieved and what are still lacking by earlier models?

**See response to comment above**

Page 3 Line 36 "builds on the code and key concepts of Dynamic TOPMODEL.": This sounds that DECIPHeR is an upgrade of Dynamic TOPMODEL. If this is the case, it is more readable to introduce the concept and formulations of Dynamic TOPMODEL first, then show the new functions and characteristics of DECIPHeR. Actually, the present form is hard to know what are same or different between two models.

**We agree this could be made clearer. We have rewritten Section 2.1 to ensure this point is clarified. As suggested by the reviewer we now introduce the key concepts of Dynamic TOPMODEL first and then make clear the changes we have made to the model code.**

Page 4 Line 18 "To realise this, DECIPHeR uses hydrological response units (HRUs)": It is hard to know whether the HRV concept has been already included in Dynamic TOPMODEL or not. I was confused similarly by many parts in this section. As mentioned earlier, please make it clear what are same or different between two models more clearly.

**See response to comment above.**

Page 6 Line 9 "In DECIPHeR, they provide the basis for river routing . . .". Ibid.

**As now made clear in Section 2.1, the river routing code is completely new so this is unique to DECIPHeR.**

Page 8 Line 12 "2.3.5 Model Structure": Unfortunately, I could hardly understand the model structure. Please describe all the equations for the terms in Figure 3 and the parameters in Table 1. At least describe where such full description of equations is available.

**We have modified this section to provide a better description of all the key equations and parameters shown in Figure 3. We have also included the derivation of the new analytical solution for the subsurface zone in the appendix (see comments in general response).**

Page 9 Line 9 "The parameter, SZM, sets..": This paragraph is particularly hard to follow. Please show the key equations how these parameters work.

**We have modified this section and included the key equations for these parameters (see response to previous comment for modifications made to the manuscript).**

Page 12 Line 44 "3.4.2 Overall model performance" and Figure 6: I am wondering why the parameters are so insensitive to the results (i.e. it is surprising that 90% of parameter sets yield NSE >0). I am also puzzled why the entire ensemble outperforms the behavioral ensemble (top 1% performance, if I understood correctly). Please elaborate these points.

**We believe that Reviewer #2 has misinterpreted parts of these results. Figure 6 shows the percentage of catchments that meet the weaker and stricter performance thresholds for each catchment. Consequently, 90% of catchments yield NSE > 0, not 90% of the parameter sets (the number of parameter sets that achieve a score of NSE > 0 varies significantly between catchments). We have modified text in section 3.4.2 to make this clearer.**

**The 'best score' from the entire ensemble for any given metric is likely to outperform the best score from the behavioural ensemble as the behavioural ensemble is the top 1% based on the combined score of the four metrics. When creating a combined score of the four metrics, you would expect some trade offs between the different metrics as any simulation is unlikely to have the best score for all four metrics.**

Page 14 Line 21 "We calculated four evaluation metrics for 10,000 model simulations for 1366 GB gauges. . .": Is this the first study to apply a hydrological to the entire Great Britain? If it is the case, clearly state so. If it is not, clearly refer the earlier efforts and compare the performance of them with this study.

**This isn't the first study to apply a hydrological model to the entirety of Great Britain. However, it is the first to have such a comprehensive model evaluation against 1,366 gauges. We have made this clearer in the discussion and included a comparison of our model performance against other GB model evaluations in Section 4.1.**

Technical comments

Page 7 Line 27 "a parameter file specifying set parameter bounds for Monte-Carlo sampling": Is "set" needed?

**Agree. We will remove 'set'.**

Page 7 Line 42 Q_SAT: I guess this term first appears. Define what this term is.

**We have modified this to "*used as the starting value for QSAT (subsurface flow)*"**

Page 12 Line 37: "13,600,600" reads 13,660,000.

**Thanks for spotting this. We have modified the text.**

Page 13 Line 4"The vast majority of gauges (90% of the whole ensemble)": 90% of the gauges or 90% of the ensemble (i.e. 9000 simulations)?

**Apologies that this was not clear. We meant 90% of the gauges and have modified the text (see response to Page 12 Line 44 above).**

Page 14 Line 27: "is" reads in.

**We have modified the text as suggested.**

Page 34 figure 6: The caption says "weaker and stricter" while the figure says "upper and lower".

**Thanks for spotting this. We have modified the legend in the figure so it says weaker and stricter thresholds.**

**DECIPHeR v1: Dynamic fluxEs and ConnectIvity for Predictions of HydRology**

Gemma Coxon[1,2], Jim Freer[1,2], Rosanna Lane[1], Toby Dunne[1], Wouter J. M. Knoben[3],
Nicholas J. K. Howden[2,3], Niall Quinn[4], Thorsten Wagener[2,3], Ross Woods[2,3]

[1]Geographical Sciences, University of Bristol, Bristol, United Kingdom, BS8 1SS
[2]Cabot Institute, University of Bristol, Bristol, United Kingdom, BS8 1UJ
[3]Department of Civil Engineering, University of Bristol, Bristol, United Kingdom, BS8 1TR
[4]Fathom Global, The Engine Shed, Station Approach, Bristol, United Kingdom, BS1 6QH

*Correspondence to:* Gemma Coxon (gemma.coxon@bristol.ac.uk)

**Abstract.** This paper presents DECIPHeR (Dynamic fluxEs and ConnectIvity for
Predictions of HydRology); a new  model framework that simulates and predicts
hydrologic flows from spatial scales of small headwater catchments to entire continents.
DECIPHER can be adapted to specific hydrologic settings and to different levels of data
availability  It is a flexible model framework which has the capability to (1)
change its representation of spatial variability and hydrologic connectivity by implementing
hydrological response units in any configuration, and (2) test different hypotheses of
catchment behaviour by altering the model equations and parameters in different parts of the
landscape. ~~and modified to represent different levels of heterogeneity, connectivity and
hydrological processes as needed.~~ It has an automated build function that allows rapid set-up
across  large model domains and is open source to help researchers and/or
practitioners use the model. DECIPHER is applied across Great Britain to demonstrate the
model framework.  It is evaluated against daily flow time series from 1,366 gauges for
four evaluation metrics to provide a benchmark of model performance. Results show the
model performs well across a range of catchment characteristics but particularly in wetter
catchments in the West and North of Great Britain. Future model developments will focus on
adding modules to DECIPHeR to improve the representation of groundwater dynamics and
human influences.

**Commented [GC1]:** RC2 Page 1 Line 15: "a new flexible model framework": Make this part more specific. What is a flexible model framework (or what is an inflexible model)?

RC2 Page 1 Line 18: "modified to represent different levels of heterogeneity, connectivity and hydrological processes as needed": Make this part more specific. All models can be "modified to represent" these in some extent. Add more concrete words in what sense DECIPHer is more adaptable compared with other models.

[revised manuscript text omitted]
.": This sounds that DECIPHeR is an upgrade of Dynamic TOPMODEL. If this is the case, it is more readable to introduce the concept and formulations of Dynamic TOPMODEL first, then show the new functions and characteristics of DECIPHeR. Actually, the present form is hard to know what are same or different between two models.

(Beven and Freer, 2001). Since its original development, and has sinceDynamic TOPMODEL has been applied in a wide range of studies (Freer et al., 2004; Liu et al., 2009, p.200; Metcalfe et al., 2017; Page et al., 2007; Younger et al., 2008) and integrated into other modelling frameworks (e.g. HydroBlocks, (Chaney et al., 2016). The core ideas of Dynamic TOPMODEL were three-fold (Beven and Freer, 2001); 1) to allow more flexibility in the definition of similarity in function for different points in the landscape, 2) to implement a non-linear routing of subsurface flow that simulates dynamically variable upslope subsurface contributing area and 3) to remain computationally efficient so that uncertainty in hydrological simulations can be estimated.

To realise this, Dynamic TOPMODEL uses hydrological response units (HRUs) to group raster-based information into non-contiguous spatial elements in the landscape that share similar characteristics (see Figure 1). Each HRU maintains hydrological connectivity in the landscape via weightings that determine the proportions of lateral subsurface flux from each HRU to all connected HRUs and flows to river cells. This solution offers key advantages in capability to traditional grid-based or lumped approaches employed by many hydrological models. Firstly, the user can split up the catchment using, for example, different landscape attributes (e.g. geology, land use) and/or spatially varying inputs (e.g. rainfall, evaporation, etc.) to define spatial similarity. This capability allows the user to modify the spatial complexity, resolution and/or hydrologic connectivity of hillslope elements and river network reaches in any configuration. Secondly, each HRU is treated as a separate functional unit in the model which can have different process conceptualisations and parameterisations. This means that more process complexity can be incorporated where needed to better suit local conditions (e.g. to account for 'point-source' human influences or more complex hydrological processes such as surface-groundwater exchanges). Finally, by grouping together similar parts of the landscape, HRUs minimise run times of the model compared to grid-based or fully distributed formulations, while still allowing model simulations to be mapped back into space.

While these key concepts that underpin Dynamic TOPMODEL address many of the challenges outlined in the introduction section, for the most part the modelit has only ever been applied to a single catchment or very simple nested gauge networkscatchments in headwater catchments basins (Peters et al., 2003). Consequently, we have completely restructured and rewritten the model code and added several new features to improve the flexibility and automation of the original Dynamic TOPMODEL code made several key advances in flexibility and in automation so the model can be applied from single small headwater catchments to regional, national and continental scales. These changes include:

1. Both legacy and new model Ccode has been updated to a FORTRAN 2003 compliant version with new array and memory handling to allowing significantly larger and more, complex gauging networks to be processed
2. The model build process is now fully automated to allow national/continental scale data to be easily and quickly processed, and to build and apply models in complex multi-catchment regions.
3. New model code and functions have been written to:
    a. Enable greater flexibility in the complexity and spatial characteristics of river network and routing properties. A newly developed river network scheme allows flow simulations to be produced for any gauged or ungauged point on a river network and segment river reaches into any length for individual hillslope-river flux contributions.

**Commented [GC5]:** RC2 Page 4 Line 18 "To realise this, DECIPHeR uses hydrological response units (HRUs)": It is hard to know whether the HRV concept has been already included in Dynamic TOPMODEL or not. I was confused similarly by many parts in this section. As mentioned earlier, please make it clear what are same or different between two models more clearly.

**Commented [GC6]:** RC1 (4) P4, L16-40: Why did the authors use HRUs instead of doing a fully-distributed model? Is it just the run time minimization? Is there a compromise in terms of adding new features such as groundwater flows and human water use? Again, I suggest adding a note on how this framework advances our capability to simulate the hydrology in comparison to numerous existing framework (see comments above)?

**Commented [GC7]:** RC2 Page 3 Line 36 "builds on the code and key concepts of Dynamic TOPMODEL.": This sounds that DECIPHeR is an upgrade of Dynamic TOPMODEL. If this is the case, it is more readable to introduce the concept and formulations of Dynamic TOPMODEL first, then show the new functions and characteristics of DECIPHeR. Actually, the present form is hard to know what are same or different between two models.

[revised manuscript text omitted]

**Commented [GC14]:** RC1 (16) Section 3.3.1: Are the precip data used here same as those shown in Fig. 3.1?

**Commented [GC15]:** RC1 (14) P9, L42: "evapotranspiration losses are highest . . ..": The figure shows PET, not the actual ET, and I believe high PET doesn't necessarily mean high ET (in water limited regimes). I think this argument is not supported unless the actual ET is shown. Could the authors clarify this?

**Commented [GC16]:** RC1 (15) P10, L32-L42: Is the river network map described consistent with the topography data described in the previous paragraph? Isn't it necessary to generate a river network map from the DEM used in the model?

[revised manuscript text omitted]

**Commented [GC17]:** RC2.  Page 12 Line 37: "13,600,600" reads 13,660,000.

**Commented [GC18]:** RC2. Page 12 Line 44 "3.4.2 Overall model performance" and Figure 6: I am wondering why the parameters are so insensitive to the results (i.e. it is surprising that 90% of parameter sets yield NSE >0). I am also puzzled why the entire ensemble outperforms the behavioral ensemble (top 1% performance, if I understood correctly). Please elaborate these points.

RC2. Page 13 Line 4"The vast majority of gauges (90% of the whole ensemble)": 90% of the gauges or 90% of the ensemble (i.e. 9000 simulations)?

high baseflow.  Flow in the Midlands region is heavily regulated by reservoirs which sustain low flows and could be a potential reason for over-estimating low flows in this area.  The bias in slope of the flow duration curve shows DECIPHeR does well at reproducing the flow variability but tends to under-estimate the slope in Scotland and North Wales suggesting that the hydrographs in these catchments are too smooth and not sufficiently flashy.

**3.4.4   Relationship Between Model Performance and Catchment Characteristics**

To further analyse and understand the reasons for good/poor model performance, relationships between key catchment characteristics and model performance were further explored.  Firstly, the catchments were grouped according to key catchment characteristics based on discharge; runoff coefficient and base flow index.  The $5^{th}$, $50^{th}$ and $95^{th}$ percentiles of NSE and RRBIAS were calculated from the ensemble of runs for all catchments within each group to explore relationships between model performance and catchment characteristics (see Table 4).  The relationship between runoff coefficient, wetness index and RRBIAS was also analysed to further explore the importance of water gains/losses on model performance.

There is a clear link between model performance and catchments with a low runoff coefficient.  Table 4 highlights poor model performance in catchments where observed runoff coefficients are less than 0.2.  In this group, the model always over-predicts (as shown by the RRBIAS result) and consequently leads to poor NSE scores.  Figure 8 shows that for many catchments where the model over-predicts flows (and particularly for catchments with a runoff coefficient less than 0.2) observed potential evapotranspiration estimates are not high enough to account for water losses culminating in an over-estimation of flows.  This is unsurprising given that currently the model maintains water balance and can't lose or gain water beyond the 'natural' conceptualisations of precipitation, discharge and evaporation dynamics.  Consequently, we are either missing a process (such as water loss due to inter-catchment groundwater flows or anthropogenic impacts) or the data is wrong.

Poorer model performance is also found in high BFI catchments (Table 4), however, the results also show we can also gain very good simulations in these types of catchments (5th percentile has a NSE score of 0.83), hence the challenge is to better understand water losses/gains in groundwater catchments  to improve the representation of groundwater dynamics in the model.

**3.4.5   Simulated Flow Time Series**

Finally, we examined the simulated flow time series for six example catchments with different characteristics.  Figure 9 shows the observed discharge, observed precipitation and the $5^{th}$-$95^{th}$ percentile uncertainty bounds of the behavioural simulations for six catchments with different characteristics (see Table 5) for a representative two-year period of the 55-year time series simulated.  The $5^{th}$-$95^{th}$ percentile uncertainty bounds are generated from the likelihood-weighted distribution of the top 1% of the model simulations using the GLUE framework (Beven, 2006).

Our results show the model can capture a range of different hydrological dynamics from wetter catchments in the North-West (Figure 9a) to drier catchments in the South-East (Figure 9b).  While model performance for groundwater catchments can be very good (Figure 9c and Table 5), it also shows that we need to incorporate additional model capability to simulate the dynamics of groundwater dominated catchments.  Where we have a very low (for Great Britain) runoff coefficient, this is assumed to involve water losses into a more regional groundwater storage not expressed at the outlet and not yet represented in this

**Commented [GC19]:** RC1  (19) Section 3.4.3 (P13, L33): The authors should present the actual time streamflow time series. Since this is the only the variable simulated/discussed, I was surprised that authors are not showing the time series plots. I suggest selecting certain representative gauging stations with varying catchment area and those located in different climatic regions for such analysis (it could be a 20 stations for example).

version of the model (Figure 9d).  While the catchments shown in Figure 9a-d are relatively un-impacted by human influences, the catchments shown in Figure 9e and 9f are heavily impacted by human influences and highlight the challenge of simulating flows nationally across catchments with diverse hydrological behaviour.

**4    Outlook and Ongoing Developments**

**4.1    National Scale Model Evaluation**

This is the first study to comprehensively benchmark hydrological model performance across GB. We calculated four evaluation metrics for 10,000 model simulations for 1,366 GB gauges to provide an initial benchmark of model performance. DECIPHeR generally performs well for the flow time series evaluated in this study, with better results in the West and North in wet catchments as compared to drier catchments in the South and East. This is a common finding for hydrological models, with many studies finding poor model performance and greatest water balance errors in drier catchments (Gosling and Arnell, 2011; McMillan et al., 2016; Newman et al., 2015; Pechlivanidis and Arheimer, 2015).  These results are also reflected in other GB model evaluation studies.  For example, (Coxon et al., 2014) applied FUSE to 24 GB catchments and found the best model performances in wet catchments compared to dry, chalk catchments, (Rudd et al., 2017) evaluated G2G for low flows across 61 GB catchments and found positive bias in low flow volumes in small catchments in the South-East of England and (Crooks et al., 2010) evaluated PDM across 120 GB catchments and found poorer model performance in groundwater dominated, drier catchments.

Poor model performance ins these catchments is partially due to some of the metrics chosen in this study, for example percent bias is most sensitive to small absolute biases in the driest catchments when compared to other metrics such as absolute bias. However, positive bias in the runoff ratio could be caused by a number of factors such as under-estimation of potential evapotranspiration (there are other UK gridded PETpotential evapotranspiration products which estimate much higher potential evapotranspiration), inter-catchment groundwater flows, and/or human influences such as water abstraction. Population density is much higher in the South and East compared to the North and West so this regional disparity in model performance could also be explained by a greater rate of abstractions and managed watercourses which alter the flow time series.  For example, 55% of the effective rainfall in the Thames catchment is licensed for abstraction (Thames Water, 2017).

These results provide an initial test of DECIPHeR capabilities against a large sample of catchments, but this is only a first-order evaluation of model performance.  A more rigorous evaluation would assess the model: over different seasons (Freer et al., 2004); under changing climatic conditions (Fowler et al., 2016); for different hydrological extremes (Coron et al., 2012; Veldkamp et al., 2018; Zaherpour et al., 2018); for multiple objectives simultaneously (Kollat et al., 2012); and, incorporate input and flow data uncertainty (Coxon et al., 2014; Kavetski et al., 2006; McMillan et al., 2010; Westerberg et al., 2016).

**4.2    Characterising Spatial Heterogeneity and Connectivity**

The intended use of DECIPHeR is to determine how much spatial variability and complexity is required for a given set of modelling objectives. It can be run as a lumped model (1 HRU), semi-distributed (multiple HRUs) or fully gridded (HRU for every single grid cell). In this paper DECIPHeR was applied across 1,366 GB gauges, with catchment masks, 5 km input grids and three classes of accumulated area and slope as classifiers for the hydrological response units, resulting in a total of 133,286 HRUs. Future work needs to consider the appropriate spatial complexity and hydrologic connectivity needed to represent relevant

**Commented [GC20]:** RC2 Page 14 Line 21 "We calculated four evaluation metrics for 10,000 model simulations for 1366 GB gauges. . .": Is this the first study to apply a hydrological to the entire Great Britain? If it is the case, clearly state so. If it is not, clearly refer the earlier efforts and compare the performance of them with this study.

**Commented [GC21]:** RC2 Page 14 Line 27: "is" reads in

**Commented [GC22]:** RC1 (22) P14, L39-45: The authors could discuss the appropriateness of different performance measures by referring some recent studies that have used a wide range of such performance measures (Veldkamp et al. 2018; Zaherpour et al. 2018). This comment is relevant to P12, L5-15 as well.

processes (Andréassian et al., 2004; Blöschl and Sivapalan, 1995; Boyle et al., 2001; Chaney et al., 2016; Clark et al., 2015; Metcalfe et al., 2015; Wood et al., 1988). While this work highlights the clear potential of a computationally-efficient large-scale modelling framework that can run large ensembles, a balance is required to ensure computational efficiency when running large ensembles that also maintains sufficient spatial complexity to represent different hydrological processes.

**4.3 Hypothesis Testing and Model Parameterisation**

To demonstrate the modelling framework we implemented a single model structure, provided in the open source model code, in all HRUs across GB and did not experiment with different model structures in different parts of the landscape. This provides a good benchmark of DECIPHeR's ability at the national scale across GB, but the results suggest different model structures are needed to represent a greater heterogeneity of hydrological responses beyond the conceptual dynamics currently implemented in this simple model (as shown in Figure 9). We can gain new process understanding of regional differences in catchment behaviour by testing different model representations (Atkinson et al., 2002; Bai et al., 2009; Perrin et al., 2001). Future work will concentrate on adding modules to DECIPHeR to enhance performance across national and continental scales with a focus on improved representation of groundwater dynamic and human influences to address poor model performance in catchments with a low runoff coefficient. Furthermore, we have ensured the code is open-source and well-documented so that the hydrological community can contribute new/different conceptualisations of the processes shown in this paper. We can gain new process understanding of regional differences in catchment behaviour by testing different model representations (Atkinson et al., 2002; Bai et al., 2009; Perrin et al., 2001).

It is challenging to parameterise a national scale hydrological model across large scales. Here we simply applied the same parameter set across each catchment. Using this basin-by-basin approach has the disadvantage of producing a "patchwork quilt" of parameter fields, with discontinuities in parameter values across catchment boundaries. This is only effective for gauged catchments (Archfield Stacey A. et al., 2015). Ongoing work aims to address these issues by implementing the multiscale parameter regionalisation (MPR) technique for DECIPHeR across GB. This technique links model parameters to geophysical catchment attributes through transfer functions applied at the finest possible resolution (Samaniego et al., 2010). The coefficients of the transfer functions are then calibrated, and parameters are upscaled to produce spatially consistent fields of model parameters at any resolution across the entire model domain. The MPR technique has been applied elsewhere, proving that it can produce seamless parameter fields across large domains and produce scale-invariant parameters (Kumar et al., 2013; Mizukami et al., 2017; Samaniego et al., 2017), which is ideal for a flexible framework such as DECIPHeR.

**5    Conclusions**

DECIPHeR is a new flexible modelling framework which can be applied from small catchment to continental scale for complex river basins resolving small-scale spatial heterogeneity and connectivity. The model is underpinned by a flexible, computationally efficient framework with a number of novel features:

1.    *Spatial variability and connectivity* - ability to modify spatial variability and connectivity in the model via the specification of hydrological response units with different topographic, landscape, input layers

**Commented [GC23]:** RC1 (24) Finally, the authors should provide caveats in the current framework and the challenges in upscaling the framework to continental and possibly to global scales. The discussion regarding advancements compared to the existing models/ongoing efforts (e.g., the National Water Model) also becomes relevant here. A note on the use on the use HRUs, and not distributed grids, should also be made.

[revised manuscript text omitted]

Commented [GC24]: RC2 Page 34 figure 6: The caption says "weaker and stricter" while the figure says "upper and lower".

[Figure]

**Figure 7.** Model performance for the best simulation (as defined by the combined rank across all four metrics) for each evaluation metric a) NSE (-), b) Bias in Runoff Ratio (%), c) Bias in Low Flow Volume (%), and d) Bias in Slope of the Flow Duration Curve between the 30th and 70th percentil

[Figure]

**Figure 8.** Scatter plot of wetness index (mean annual precipitation divided by mean annual potential evapotranspiration), runoff coefficient (mean annual discharge divided by mean annual precipitation) and bias in runoff ratio for each GB catchment evaluated in this study. Any points above the horizontal dotted line are where runoff exceeds total rainfall inputs in a catchment and any points below the curved line are where runoff deficits exceed total potential evapotranspiration in a catchment.

[Figure]

**Figure 9.** Observed discharge and uncertainty bounds for the behavioural simulations (5[th] and 95[th] percentile of the likelihood-weighted simulated discharge) for six catchments with different characteristics (shown in Table 5). The plots show a two year period (2010-2012) from the 55 year time series simulated.

**Commented [GC25]:** RC1 (19) Section 3.4.3 (P13, L33): The authors should present the actual time streamflow time series. Since this is the only the variable simulated/discussed, I was surprised that authors are not showing the time series plots. I suggest selecting certain representative gauging stations with varying catchment area and those located in different climatic regions for such analysis (it could be a 20 stations for example).

---

## Author Response (AR2)

**GMD Submission by Coxon et al**
**DECIPHeR v1: Dynamic fluxEs and ConnectIvity for Predictions of HydRology**

**General Response**

We thank the reviewers for taking the time to review the revised paper and their comments.

Detailed responses to all comments are provided below. Author responses are in **bold** and any modifications to the manuscript are in *italic* below each of the reviewer's comments.

10 Gemma Coxon, April 2019

**Reviewer #1**

The authors have addressed many of the major concerns raised during the previous review, but not all to the satisfactory level. I also have a few additional comments for the authors
15 which need to be fully addressed (listed below).

(1) For many comments, the authors have just provided a response and it appears that they have not included a clarification in the revised version of the manuscript (I assume this because they did not include a reference to the manuscript in their response). These include Reviewer 1, Comments 5, 8, 18 etc. Please ensure that these have been addressed in the
20 revised manuscript.

**We have now included a clarification in the revised version of the manuscript for Reviewer 1 Comments 5 and 8. We will not be adding additional text in response to Comment 18 (see response to comment 2 below).**

**Comment 5**

25 *The DEM must contain no sinks or flat areas to ensure that the river network and catchments can be properly delineated as is common in digital terrain analyses. This means that any real inland sinks (such as lakes) will be filled. Accounting for these features in the modelling framework will be a focus for future model development.*

**Comment 8**

30 *If a gauge does not have an initial flow, then the initial flow is either calculated from the mean of the data or set to a value of 1mm/day (as a representative starting flow for most catchments) if no flow data is available.*

(2) Comment 18 in the previous review: what are the models that directly use PET as input without simulating it? This sounds like a really crude way to do hydrological modeling, i.e.,
35 using PET as input and estimating ET unless the authors meant otherwise. Please include examples in the revised version and provide further clarifications.

We simply do not understand why the reviewer is still requiring clarification on this matter. Using PET as input is not crude, it is perfectly accepted practice and we should not need to justify the inputs for many conceptual modelling approaches. Some of the most used hydrological modelling systems in the world use potential evaporation (PET) as an input (i.e. TOPMODEL (e.g. Beven, 1997) that has reputedly the largest ever distribution of applications and published journal papers - some >400 papers).

There are many variants of how you can conceptualise the actual evaporation usage in the model just as there are many variants of how you might formulate PE in the first instance depending on the level of data available and assumptions (i.e. Penman/Penman-Monteith/Hamon/Thornthwaite/Priestly-Taylor/etc. - see Federer et al., 1996). Models that are also fully distributed use PET, such as some of those involved in the Distributed Modelling Intercomparison Project (i.e. Smith et al., 2004); as does the well known PDM model (Calver et al., 2001); and another fully distributed model Grid-2-Grid (Bell et al., 1998; 2007).

This is simply a choice and is very much in keeping with hydrological modelling practice, we do not feel the need to deal with this comment further and have made clear in the manuscript what we use as an input into our model framework and the equation used to calculate Actual Evapotranspiration (equation 2).

Bell, V.A. and Moore, R.J., 1998. A grid-based distributed flood forecasting model for use with weather radar data: Part 1: Formulation. Hydrol. Earth Syst. Sci., PD 265-281.

Bell, V.A., Kay, A.L., Jones, R.G. and Moore, R.J. (2007) Use of a grid-based hydrological model and regional climate model outputs to assess changing flood risk. International Journal of Climatology 27(12), 1657-1671.

Beven, K.J. (1997) TOPMODEL: A critique. Hydrological Processes 11(9), 1069-1085.

Calver, A., Lamb, R., Kay, A.L. and Crewett, J., 2001. The continuous simulation method for river flood frequency estimation. Department for Environment, Food and Rural Affairs Project FD0404 Final Report, CEH Wallingford, UK. 56pp + appendices.

Federer, C.A., Vorosmarty, C. and Fekete, B., 1996. Intercomparison of methods for calculating potential evaporation in regional and global water balance models, WRR 32(7): 2315-2321.

Smith, M.B., Georgakakos, K.P. and Liang, X. (2004) The distributed model intercomparison project (DMIP). Journal of Hydrology 298(1-4), 1-3.

(3) Figure 9: I suggest adding a line for mean and then doing the shading for 5-95th percentile. Also, please include some statistics such as RMSE, R2, and bias to quantify the uncertainty in the simulations.

Thanks for the suggestion – we have added a line for the median to these plots alongside the 5th-95th percentile (please see the revised paper for the new plot).

The metrics that the reviewer suggests do not quantify the uncertainty in the simulations – they evaluate the performance of the model. We have already included performance metrics (including the bias and NSE scores) for the mean of the ensemble and the best simulation in Table 5 which accompanies Figure 9.

(4) Comment #20 in the previous review: "outside the scope of this paper". Why is model validation exercise outside of scope of a model development paper? I believe, that is the

purpose of the paper because the paper is not driven by science questions. It is purely a model description and validation. Right? I agree that some validation is provided, but given that the bias is large for most locations (Figures 7 and 9), further investigation is warranted.

**As stated in our previous response, we have already undertaken an extensive model**
5 **evaluation by evaluating the model at 1,366 gauges for four different metrics. We have produced a plot (as suggested by Reviewer #1) that shows the observed and simulated flow for different flow percentiles (Q5, Q50 and Q95). We will include this plot and the following text in the supplementary material for the paper.**

*In this supplementary document we provide additional analysis of the national-scale model*
10 *simulations described in 'DECIPHeR v1: Dynamic fluxEs and ConnectIvity for Predictions of HydRology', Coxon et al.*

*Figure S1 shows the ability of the model to simulate observed flow percentiles (Q95, Q50 and Q5). The results show that for many of the catchments the model can capture these flow percentiles. However, the model tends to overpredict the flow percentiles (particularly for*
15 *Q5) in drier catchments where runoff is low. These results are consistent with the model results presented in the main paper.*

[Figure]

*Figure S1. Observed and simulated flow percentiles for each gauge. The red scatter point signifies the median value of the simulated flow percentiles from the behavioural simulations,*
20 *while the black line shows the $5^{th}$-$95^{th}$ percentile of the simulated flow percentiles from the behavioural simulations*

(5) Minor editorial issue; Abstract, L18: I think something like the following reads better: "which includes the capability to (1) change the representation of …" because the model doesn't do things, it represents certain parameterizations and capabilities.

**We agree and have changed this sentence in the manuscript.**

**Reviewer #2**

I have read through the revised manuscript. I am convinced that the two issues I raised for the earlier version have been fully addressed. Now the characteristics and strengths of the model are clearly depicted in Introduction Section. The model description part has been very much streamlined highlighting the advances in modeling. The discussion part concisely highlights the novelty and significance in the simulation (Section 4.1). I recommend this manuscript be published after revisiting the following points. Hope this model will be widely accepted by the hydrology community.

**We thank the reviewer for their kind comments.**

Page 8 Line 17 "subsurface store… if it is also full". Does subsurface store have a certain capacity? How was it set in this study (I cannot find this in Table 1)?

**The capacity of the subsurface store is determined by Smax.  We have re-written this sentence to clarify this point.**

*Once the root zone reaches maximum capacity (i.e. deficit of zero and conceptually analogous to field capacity), any excess rainfall input that is remaining is added to the unsaturated zone ($S_{uz}$) where it is routed to the saturated zone ($S_D$). If the saturated zone is also full (as determined by $S_{max}$), $Q_{EXUS}$ is added to the saturation excess storage ($S_{EX}$) and routed directly overland as saturated excess overland flow ($Q_{OF}$).*

Page 8 Line 26 "The dryness of the saturated zone is represented by the storage deficit": It sounds a bit odd to me because saturated zone cannot be dry. Although I speculate this expression is linked to the fundamental concept of the TOPMODEL, possibly the authors will be able to find some better ones

**We agree that this sentence is a little confusing and we have removed it from the manuscript.**

**DECIPHeR v1: Dynamic fluxEs and ConnectIvity for Predictions of HydRology**

Gemma Coxon[1,2], Jim Freer[1,2], Rosanna Lane[1], Toby Dunne[1], Wouter J. M. Knoben[3],
Nicholas J. K. Howden[2,3], Niall Quinn[4], Thorsten Wagener[2,3], Ross Woods[2,3]

[1]Geographical Sciences, University of Bristol, Bristol, United Kingdom, BS8 1SS
[2]Cabot Institute, University of Bristol, Bristol, United Kingdom, BS8 1UJ
[3]Department of Civil Engineering, University of Bristol, Bristol, United Kingdom, BS8 1TR
[4]Fathom Global, The Engine Shed, Station Approach, Bristol, United Kingdom, BS1 6QH

*Correspondence to:* Gemma Coxon (gemma.coxon@bristol.ac.uk)

**Abstract.** This paper presents DECIPHeR (Dynamic fluxEs and ConnectIvity for Predictions of HydRology); a new model framework that simulates and predicts hydrologic flows from spatial scales of small headwater catchments to entire continents. DECIPHeR can be adapted to specific hydrologic settings and to different levels of data availability. It is a flexible model framework which has the capability to (1) change its representation of spatial variability and hydrologic connectivity by implementing hydrological response units in any configuration, and (2) test different hypotheses of catchment behaviour by altering the model equations and parameters in different parts of the landscape. It has an automated build function that allows rapid set-up across large model domains and is open source to help researchers and/or practitioners use the model. DECIPHeR is applied across Great Britain to demonstrate the model framework. It is evaluated against daily flow time series from 1,366 gauges for four evaluation metrics to provide a benchmark of model performance. Results show the model performs well across a range of catchment characteristics but particularly in wetter catchments in the West and North of Great Britain. Future model developments will focus on adding modules to DECIPHeR to improve the representation of groundwater dynamics and human influences.

**Commented [GC1]:** Reviewer #1

(5) Minor editorial issue; Abstract, L18: I think something like the following reads better: "which includes the capability to (1) change the representation of …" because the model doesn't do things, it represents certain parameterizations and capabilities.

[revised manuscript text omitted]

**Commented [GC2]:** Reviewer #1

For many comments, the authors have just provided a response and it appears that they have not included a clarification in the revised version of the manuscript (I assume this because they did not include a reference to the manuscript in their response). These include Reviewer 1, Comments 5, 8, 18 etc. Please ensure that these have been addressed in the revised manuscript.

[revised manuscript text omitted]

**Commented [GC3]:** Reviewer #1

(1) For many comments, the authors have just provided a response and it appears that they have not included a clarification in the revised version of the manuscript (I assume this because they did not include a reference to the manuscript in their response). These include Reviewer 1, Comments 5, 8, 18 etc. Please ensure that these have been addressed in the revised manuscript.

**2.3.4 Model Structure**

The description below details the model structure that is provided in the open source code (see Figure 3 and Table 1). While the code is built to be modular and extensible so that a user can easily implement multiple model structures if so wished, the aim of this paper and the initial focus of the code development was on applying the model across large scales and beginning with a release that has relatively simple representations of the core processes. Thus, we provide a single model structure in the open source code that serves as a model benchmark to be built upon in future iterations.

The model structure consists of three stores defining the soil profile ($S_{RZ}$, $S_{UZ}$, $S_D$ in Figure 3), which are implemented as lumped stores for each HRU. The first store is the root zone storage ($S_{RZ}$). Precipitation ($P$) is added to this store and then evapotranspiration ($ET$) is calculated and removed directly from the root zone. The maximum specific storage of $S_{RZ}$ is determined by the parameter $SR_{max}$. Actual evapotranspiration from each HRU depends on the potential evapotranspiration ($PET$) rate supplied by the user and the root zone storage using a simple common formulation where evapotranspiration is removed at the full potential rate from saturated areas (i.e. if the root zone storage is full) and at a rate proportional to the root zone storage in unsaturated areas:

$$ET = PET * (S_{RZ}/SR_{max})$$

**Equation 2**

Once the root zone reaches maximum capacity (i.e. deficit of zero and conceptually analogous to field capacity), any excess rainfall input that is remaining is  added to the unsaturated zone ($S_{uz}$) where it is routed to the saturated zone ($S_D$) If the saturated zone is also full (as determined by $S_{max}$), $Q_{EXUS}$ is added to the saturation excess storage ($S_{EX}$) and routed directly overland as saturated excess overland flow ($Q_{OF}$). The unsaturated zone links the $S_{RZ}$ and saturated zones according to a linear function that includes a gravity drainage time delay parameter ($Td$) for vertical routing through the unsaturated zone. The drainage flux ($Q_{uz}$) from the unsaturated zone to the saturated zone is at a rate proportional to the ratio of unsaturated zone storage ($S_{uz}$) to storage deficit ($S_D$):

$$Q_{UZ} = S_{UZ}/(S_D * Td)$$

**Equation 3**

 Changes to storage deficits for each HRU are dependent on recharge from $S_{UZ}$ ($Q_{UZ}$), fluxes from upslope HRUs ($Q_{IN}$) and downslope flow out of each HRU ($Q_{SAT}$), with subsurface flows for each HRU distributed according to the DTA flow distribution matrix described in section 2.2.4.

$$\frac{dS_D}{dt} = Q_{SAT} - Q_{IN} - Q_{UZ}$$

**Equation 4**

Where $S_D$ is the current deficit in the saturated zone, $Q_{SAT}$ is outflow from this HRU, $Q_{IN}$ is inflow into the HRU representing subsurface flow from other HRUs  and $Q_{UZ}$ is inflow into the HRU representing drainage from the unsaturated zone of this HRU. This equation is solved sequentially for each HRU and provides values for the deficit $S_D$ and outflow $Q_{SAT}$ at time step $t$ for each HRU. In DECIPHeR, this equation is solved analytically

**Commented [GC4]:** Reviewer #2

Page 8 Line 17 "subsurface store… if it is also full". Does subsurface store have a certain capacity? How was it set in this study (I cannot find this in Table 1)?

**Commented [GC5]:** Reviewer #2

Page 8 Line 26 "The dryness of the saturated zone is represented by the storage deficit": It sounds a bit odd to me because saturated zone cannot be dry. Although I speculate this expression is linked to the fundamental concept of the TOPMODEL, possibly the authors will be able to find some better ones

[revised manuscript text omitted]